# MiniCPM: Unveiling the Potential of Small Language Models with Scalable Training Strategies

**Shengding Hu**[1], **Yuge Tu**[2], **Xu Han**[1,3]*, **Chaoqun He**[1], **Ganqu Cui**[1], **Xiang Long**[2],
**Zhi Zheng**[2], **Yewei Fang**[2], **Yuxiang Huang**[1], **Weilin Zhao**[1], **Xinrong Zhang**[1],
**Zheng Leng Thai**[1],**Kaihuo Zhang**[2], **Chongyi Wang**[2], **Yuan Yao**[1],
 **Chenyang Zhao**[1], **Jie Zhou**[2], **Jie Cai**[2], , **Zhongwu Zhai**[2], **Ning Ding**[1],
**Chao Jia**[2], **Guoyang Zeng**[2], **Dahai Li**[2], **Zhiyuan Liu**[1]*, **Maosong Sun**[1]*
[1]Department of Computer Science and Technology, Institute for Artificial Intelligence,
  Beijing National Research Center for Information Science and Technology,
  Tsinghua University, Beijing, China.
[2]Modelbest Inc.
[3]Shanghai Artificial Intelligence Laboratory.
shengdinghu@gmail.com, {hanxu2022,liuzy,sms}@tsinghua.edu.cn

## Abstract

The burgeoning interest in developing Large Language Models (LLMs) with up to trillion parameters has been met with concerns regarding resource efficiency and practical expense, particularly given the immense cost of experimentation. This scenario underscores the importance of exploring the potential of Small Language Models (SLMs) as a resource-efficient alternative. In this context, we introduce MiniCPM, specifically the 1.2B and 2.4B non-embedding parameter variants, not only excel in their respective categories but also demonstrate capabilities on par with 7B-13B LLMs. While focusing on SLMs, our approach exhibits scalability in both model and data dimensions for future LLM research. Regarding model scaling, we employ extensive model wind tunnel experiments for stable and optimal scaling. For data scaling, we introduce a Warmup-Stable-Decay (WSD) learning rate scheduler (LRS), conducive to continuous training and domain adaptation. We present an in-depth analysis of the intriguing training dynamics that occurred in the WSD LRS. With WSD LRS, we are now able to efficiently study data-model scaling law without extensive retraining experiments on both axes of model and data, from which we derive the much higher compute optimal data-model ratio than Chinchilla Optimal. Additionally, we introduce MiniCPM family, including MiniCPM-DPO, MiniCPM-MoE and MiniCPM-128K, whose excellent performance further cementing MiniCPM's foundation in diverse SLM applications. MiniCPM models are available publicly [1].

## 1 Introduction

Following the revelation of the scaling law (Kaplan et al., 2020), there has been a vigorous pursuit in the field of Large Language Models (LLMs) (Hoffmann et al., 2022; Bai et al., 2023; Gemini et al., 2023; Chowdhery et al., 2023; Achiam et al., 2023), encompassing models with up to an astonishing number of parameters in the trillions (Fedus et al., 2022). These models have emerged as a pivotal driving force in the evolution of artificial intelligence.

Nonetheless, the training of such large-scale models is both financially burdensome and operationally inefficient. On one hand, the empirical understanding of the mechanisms underpinning the training of LLMs remains elusive. Given the significant economic and

---

*Corresponding Authors.
[1]https://github.com/OpenBMB/MiniCPM

environmental costs, experiments on LLMs are prohibitively expensive for most researchers and corporations. On the other hand, the deployment of these colossal models in everyday scenarios, such as on personal computers or smartphones, is either inefficient or unfeasible. Both aspects underscore the imperative to refocus efforts on comprehensively exploring smaller, yet potent, language models (SLMs). These models on the one hand provide efficient solutions to practical deployment, on the other hand, if trained with scalable strategies, they can potentially guide the development of future larger models.

Recently, a resurgence of interest has been observed in the domain of SLMs, evidenced by the advent of a series of innovative models such as the Phi series (Gunasekar et al., 2023; Li et al., 2023b; Javaheripi & Bubeck, 2023), TinyLlama (Zhang et al., 2024a), MobileLLM (Liu et al., 2024), and Gemma (Banks & Warkentin, 2024), among others. While these models have significantly contributed to the expansion and diversification of the SLM landscape, there remain two pivotal areas where these models have yet to fully satisfy prevailing interests: (1) the development of comprehensive abilities akin to those exhibited by LLMs; and (2) the formulation of transparent and scalable training methodologies that could further propel the evolution of both SLMs and LLMs.

In this paper, we introduce MiniCPM, a series of SLMs, which primarily builds on two models, endowed with 2.4B and 1.2B non-embedding parameters respectively, and they rank preeminently in their respective 2B and 1B scale categories. MiniCPM also exhibits comparable capabilities to those of 7B~13B language models, such as Llama2-7B (Touvron et al., 2023), Mistral-7B (Jiang et al., 2023), Gemma-7B (Banks & Warkentin, 2024), and Llama-13B (Touvron et al., 2023), etc. Notwithstanding their small model sizes, our training methodology is meticulously designed to facilitate seamless scaling of both model scale and data horizons. This is exemplified through our model wind tunnel experiments that encompass comprehensive hyper-parameter optimization (Section 3), and the deployment of a WSD (Warmup-Stable-Decay) learning rate scheduler (Section 4). The latter is tailored for continuous training with an un-predefined pre-training token number and makes the reusing of model intermediate checkpoints highly feasible. A detailed analysis of the training dynamics of MiniCPM is presented, suggesting that the WSD scheduler demonstrates the intriguing loss landscape of model pre-training. With the WSD scheduler, we are now also capable of studying the data-model scaling law with linear effort on the model axis and a negligible effort on the data axis, while the traditional ones need quadratic effort considering the scaling along both model and data axes. The result of the scaling law indicates a much higher data size/model size ratio compared with Chinchilla Optimal (Hoffmann et al., 2022).

Moreover, we introduce the MiniCPM family, including MiniCPM-DPO, MiniCPM-128K, and MiniCPM-MoE. We conduct evaluations of the MiniCPM family against established benchmarks and illuminate their impressive capabilities as SLMs: (1) The foundation models surpass Mistral-7B, and LLama-13B. (2) The DPO model surpasses zephyr-7B (Tunstall et al., 2023) on MTBench (Zheng et al., 2024) (3) The 2.4B MiniCPM-128K model demonstrates performance either surpassing or matching that of models like Yarn-Mistral-7B-128K (Peng et al., 2023) and ChatGLM3-6B-128K (Du et al., 2021). (4) The MiniCPM-MoE, with 4B activated parameters, is on par with Llama2-34B (Touvron et al., 2023).

In summary, MiniCPM propounds a new stage in the development of small language models, exemplifying the latent potential within SLMs and advocating for a more scientific and sustainable approach toward scaling up LLMs.

## 2 Related Work

**Small Language Models.** "Small Language Models" (SLMs) is an evolving concept that has undergone significant transformations over time. Presently, SLMs are generally construed as models that are smaller in scale compared to the well-known LLMs, typically not exceeding 7 billion parameters. These models are distinguished by their capacity for deployment on end-user devices, such as personal computers and smartphones, even in the absence of a GPU. Notable examples within the current landscape of SLMs include the Phi series (Gunasekar et al., 2023; Li et al., 2023b; Javaheripi & Bubeck, 2023), TinyLlama (Zhang

et al., 2024a), MobileLLM (Liu et al., 2024), and Gemma (Banks & Warkentin, 2024), etc. A variety of methodologies have been explored to augment the efficacy of SLMs. These include the incorporation of high-quality data (Gunasekar et al., 2023; Li et al., 2023b; Javaheripi & Bubeck, 2023), the application of structure pruning techniques (Xia et al., 2023), and the reconfiguration of model architectures (Liu et al., 2024), among others. MiniCPM enhances the capabilities of SLMs through a meticulous amalgamation of hyper-parameter optimization, strategic training methodologies, architectural design, and high-quality data.

**Scalable Pre-training Strategies.** Since the discovery of scaling law (Kaplan et al., 2020; Rae et al., 2021; Aghajanyan et al., 2023), scientifically and predictably (Achiam et al., 2023; Hu et al., 2023; Du et al., 2024) scaling up the LLMs has been pursued from diverse perspectives, especially for the pre-training stage. In terms of training stability, the Tensor Program series (Yang et al., 2022; 2023) is introduced to ensure optimal hyper-parameter consistency across varying model scales, a technique employed in training CerebrasGPT (Dey et al., 2023). Furthermore, Wortsman et al. (2023) suggest leveraging smaller models to anticipate and mitigate instabilities in larger model training. From the training data standpoint, various data-centric strategies have been advocated (Xie et al., 2024; Shi et al., 2023; Ye et al., 2024). In the realm of training methodologies, prior research has delved into diverse learning rate schedulers (LRS) (Howard & Ruder, 2018; Raffel et al., 2020; Hundt et al., 2019), with the Cosine LRS (Loshchilov & Hutter, 2016) emerging as the predominant choice in LLMs. Kaplan et al. (2020) and Hoffmann et al. (2022) have meticulously examined the hyper-parameters of Cosine LRS, thereby laying a foundational groundwork for subsequent pre-training works. Of these, DeepSeek (Bi et al., 2024) bears the closest resemblance to our proposed WSD LRS. Concerning batch size scheduling, Smith et al. (2017) advocates for incrementing batch size as an alternative to diminishing learning rate, a strategy recently adopted by Yi-9B (Young et al., 2024).

## 3 Model Wind Tunnel Experiments

Although we target at training SLMs that can be quickly deployed onto end devices, we envision that many aspects of model training are universal across scales. Extensive experiments should be conducted through an SLM to explore the limit of SLM before transferring the experience into LLMs. These experiments take the spirit of wind tunnel testing in developing an aircraft, thus we name it Model Wind Tunnel Experiments (MWTE). In this paper, the MWTE contains three parts: (1) Hyper-parameters; (2) Optimal Batch-size Scaling; and (3) Optimal Learning Rate Stability.

### 3.1 Scaling Hyper-parameters Invariant LM

Hyper-parameters have a significant impact on the performance of a model. However, adjusting hyper-parameters for each model in traditional training is not feasible for LLMs. Even for SLM like MiniCPM, extensive experiments on hyper-parameters search take a lot of resources. Tensor Program (Yang et al., 2022; 2023) proposes a framework to stabilize the hyper-parameters for models with different scales. The main part of the Tensor Program is the width scaling (Yang et al., 2022) and the depth scaling (Yang et al., 2023). The former technique supports CerebrasGPT (Dey et al., 2023) to predict the loss of LLMs more accurately. In MiniCPM, we use both two scaling techniques. The specific scaling operations are listed in Table 3. We do not apply the attention softmax scaling techniques proposed in (Yang et al., 2022) (See Table 3 of their first version). Despite Yang et al. (2023) observing that depth scaling for a network with block depth larger than two is not satisfying, we find the resulting optimal learning rate is stable empirically. Details of the hyper-parameters and Tensor Program Operations are in Appendix A.1.

### 3.2 Optimal Batch Size

Batch size determines the balance between the convergence speed of the model and the consumption of computational resources. If the batch size is too large, it will result in a significant amount of data and computation costs. On the other hand, if the batch size

is too small, it will require a large number of training steps and may result in a limited decrease in the loss function. We follow Kaplan et al. (2020) to determine the batchsize from expected loss, with a slight modification from their setting (see Appendix A.2). We conduct experiments on 0.009B, 0.03B, and 0.17B models, respectively, toward this goal. Each model size is trained on 6 batch sizes with a global learning rate of 0.01 and cosine learning rate scheduler. We observe the trend of the optimal batch size with loss on the C4 (Raffel et al., 2019) dataset (red line in the Figure 1).

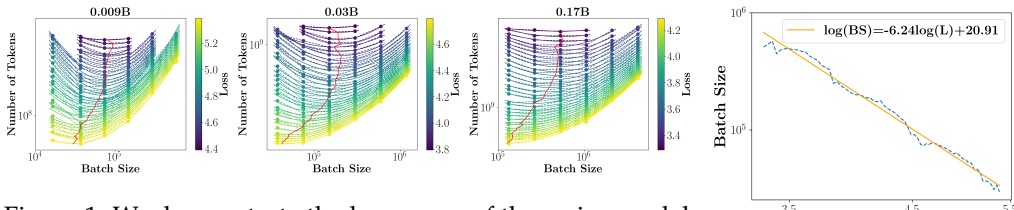

Figure 1: We demonstrate the loss curve of three size models trained using different batch sizes. Each vertical line formed by points with a gradient color represents a training curve. Lighter colors denote higher loss.

Figure 2: The connected optimal batch sizes.

As shown in Figure 1, we plot the batch size in the x-axis, and token consumption in the y-axis, the color of the points represents a loss. Thus a vertical line formed by the color points denotes a training curve. we use parabolas to fit the equal-loss points and connect the minima of the parabolas with red lines. The lines demonstrate the optimal batch size shifts large as the loss decreases. We then connect the three lines (see Figure 2) and find that the lines connect each other well into a linear relationship in the log space, from which we obtain the following relationship between batch size $bs$ and C4 Loss $L$: $bs = \frac{1.21 \times 10^9}{L^{6.24}}$. We note that it might seem strange that the batch size should be estimated from a rough loss prediction that we can only have after training. We provide our comment in Appendix A.2.

### 3.3 Optimal Learning Rate

Due to our use of Tensor Program (Yang et al., 2022; 2023), we anticipate that the learning rate, will not undergo significant changes during model scaling. To verify this, we conduct six sets of learning rate experiments at 0.04B, 0.1B, 0.3B, and 0.5B. In Figure 3, we find that although the model size has increased by ten times, the optimal base learning rate [2] does not show a noticeable shift and remains around 0.01. We further conduct a simple validation on a scale of 2.1B and confirm that a learning rate of 0.01 indeed achieves the lowest loss.

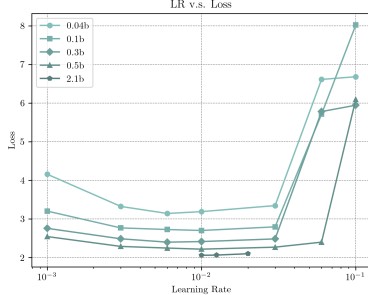

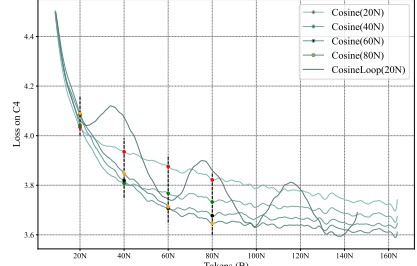

Figure 3: Loss vs Learning Rate. After applying for the Tensor Program, the learning rate shift becomes minimal.

Figure 4: Cosine Learning Rate Scheduler with different periods. The Y-axis is the loss on the C4 corpus.

---

[2]The actual learning rate of 2-D tensors will be scaled according to Tensor Program.

## 4 WSD Learning Rate Scheduler

### 4.1 Analysing Cosine LRS

The learning rate scheduler (LRS), which adjusts the learning rate used in different stages of training, is crucial for model performance. The current commonly used learning rate strategy is the Cosine LRS (Kaplan et al., 2020; Hoffmann et al., 2022; Rae et al., 2021; Touvron et al., 2023; Bai et al., 2023; Almazrouei et al., 2023), which gradually decreases the learning rate following a cosine curve after it reaches its maximum after the warmup stage.

A key hyper-parameter in the Cosine LRS is the step $T$ at which Cosine decreases to the minimum for the first time. Typically, $T$ is set to the total training step $S$ for training with a predefined training step. Generally, it is believed that the learning rate should be high to enable sufficient exploration. For example, Kaplan et al. (2020) demonstrate that the loss decreases when the summed learning rate over the entire training increases (see Figure 22 in their paper). This indicates setting $T < S$ is not optimal. On the other hand, Hoffmann et al. (2022) make a key observation that setting $T > S$ results in dropped performance while setting $S = T$ results in improved training efficiency, confirming that the learning rate shouldn't be kept high throughout the training. To reproduce these observations, we conduct experiments on the 0.036B model. We try $Cosine(T)$ and $CosineLoop(T)$ LRS, following the formula shown in Appendix B.1. The result can be seen in Figure 4. We can see that when the training step is $S = 20N, 40N, 60N, 80N$, the lowest loss is always achieved by the $Cosine(T)$ where $T = S$. Both $T < S$ and $T > S$ are not optimal.

We hypothesize that the Cosine LR performs exceptionally well when $T = S$ because of the following two reasons: (1) Cosine LRS with $T = S$ has a longer duration of *high learning rate* training compared to $T < S$ and other LRS such as Linear LRS. This high learning rate might help the model find a better global optimum. (2) Cosine LRS with $T = S$ has a more thorough learning rate decay phase compared to Cosine LRS with $T > S$ and Constant LRS. This learning rate decay may involve unique training dynamics that enable the model to find a better local optimum.

### 4.2 WSD LRS

In light of the above perspective, we propose to explicitly divide the training stage into the high learning rate stage and learning rate decay stage. We name it as the Warmup-Stable-Decay (WSD) LRS. Especially, the WSD LRS contains three stages: the warmup stage (whose end step is denoted by $W$), the stable training stage (whose end step is denoted by $T$), and the remaining decay stage. The function form of WSD is:

$$WSD(T; s) = \begin{cases} \frac{s}{W}\eta, & s < W \\ \eta, & W < s < T \\ f(s - T)\eta, & T < s < S \end{cases} \tag{1}$$

where $0 < f(s - T) \leq 1$ is a decreasing function about $s$, $\eta$ is the maximum learning rate. Typically, as long as the warmup stage is enough, it affects little performance, therefore, we omit $W$ in the subsequent discussion. With an abuse of notation, we will denote WSD with a clear stop point

### 4.3 Experiments

Next, we present several experimental findings of WSD LRS.

**Loss Decreases Dramatically in Decay Stage.** We try WSD LRS on 0.036B models. As shown in Figure 5, in the decay stage, as the learning rate begins to decrease, the loss experiences a significant rapid decline and quickly decreases to be equal to or lower than the Cosine LRS at step $T = S$. At the same time, we can reuse the model before decay and continue training with the previous high learning rate. After more steps of training $S'$, we can also perform annealing to achieve the same loss as the Cosine LRS at $Cosine(S')$. This

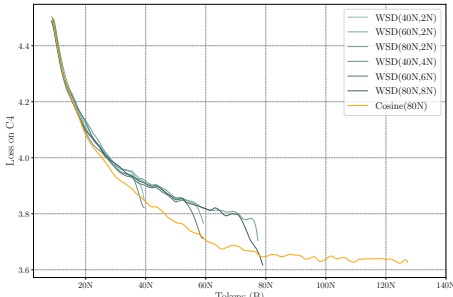

Figure 5: Model training loss has a sudden decrease in the decay stage of WSD LRS.

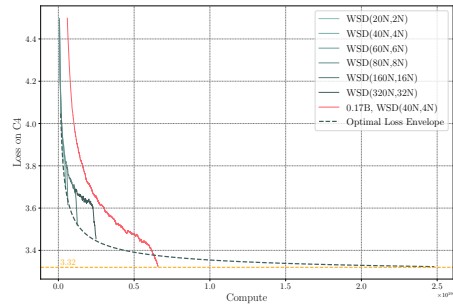

Figure 6: Continous training a 0.036B model can match the performance of 0.17B model with an acceptable increase in training compute.

verifies our assumption that the training stage can be explicitly split into the stable training and decay stages.

**10% Steps are Enough.** From the two-stage training perspective, shortening the decay stage will greatly benefit the fast test of different model checkpoints of stable training. Therefore, we conduct experiments that start from the same stable training checkpoints and have different decay steps. Also shown in Figure 5, among all three stable training checkpoints in 40N, 60N, and 80N training data, having a decay of 10% of the total tokens is sufficient to achieve the best results, while a decay of 2.5% of total tokens falls short. Therefore, in the subsequent training experiments, we use a decay of about 10% to ensure full convergence.

**Effective Data Scaling with WSD LRS.** With WSD LRS, we can continuously train the LM to extreme convergence. To further demonstrate the potential of training a fixed-sized model to convergence, we compare continuously training a 0.036B LM with a 0.17B model with 40N data. In Figure 6, the green lines represent 0.036B models trained with different stable training tokens. Despite the last point of the 0.036B series being trained with many more tokens than Chinchilla Optimal (Hoffmann et al., 2022), it still has space for performance improvement.

To find the limit of continuously training this fixed-sized LM, we estimate how the model's optimal performance changes with its computation during continuous training. By optimal performance, we mean the loss of training token $D$ is achieved by $WSD(D, 0.1D)$. With a series of $D$, the losses will form the optimal loss envelope. Due to uncertainty about the function form of the loss envelope, we try two fitting formulas: (1) exponential: $L(C) = \alpha e^{-\beta C} + L_0$ and (2) power-law: $L(C) = \beta C^{-\alpha} + L_0$. The fitting results for both functions are in Appendix B.2. We find that the power-law form fits better (similar to the Cosine LRS (Kaplan et al., 2020)). In Figure 6, the fitted curve is shown in green dotted lines. To intuitively estimate and comprehend the effect of continuous training such a fixed-sized model, we also trained a 0.17B model with $WSD(40N, 4N)$, which is shown in pink in Figure 6. We can see that a 0.036B model can match the performance of a 0.17B model with an acceptable increase ($\sim$ 4 times) in training compute while saving a lot of inference computation (Sardana & Frankle, 2023) (saving $\sim$ 5 times per inference call), indicating a better inference-compute-optimal setting (Sardana & Frankle, 2023).

> **Note**
>
> To deepen our understanding of WSD scheduler, we provide a detailed analysis of the model training dynamics in Appendix D. The WSD scheduler also helps us estimating the scaling law better and more efficiently, which can be found in Appendix E.

## 5 Two Stage Pre-training Strategy

Typically, the training of instruction following LLMs contains the pre-training stage and the supervised fine-tuning (SFT) stage (Zhang et al., 2023; Wei et al., 2021). In the pre-training stage, the data is composed of large-scale unlabeled data, while in the SFT stage, high-quality labeled data becomes the optimization target. In light of the pronounced loss decrease observed during the decay stage of the WSD LRS, we postulate that the integration of high-quality labeled data in this phase presents dual advantages:

- Introducing this data during the annealing phase, in addition to the SFT stage, fosters a more comprehensive model learning. Specifically, it facilitates a more pronounced loss reduction in relation to the SFT data distribution, rather than the pre-training data distribution. This approach is more congruent with actual user scenarios.
- In contrast to a uniform distribution of high-quality data throughout the entire pre-training process, this method enhances training by concentrating on data and sustaining continuous pre-training. If we do not predetermine a training step, we will repeat a small dataset throughout an ongoing pre-training process, which could lead to negative effects.

Based on these two hypotheses, we propose the following training strategy: during the pre-training phase, only use large-scale coarse-quality pre-training data, which is abundant and can support continuous training when provided with more computational resources. During the annealing phase, we use diverse and high-quality knowledge and ability-oriented SFT data, mixed into the pre-training data.

To validate the advantages of our training strategy, we conduct comparison experiments using (A) MiniCPM-2.4B's intermediate checkpoint in the stable stage; and (B) MiniCPM-1.2B's last checkpoints in the stable stage. Specifically, we compare the following:

1. A-1: 2.4B model, decay using only pre-training data, followed by 4B token SFT.
2. A-2: 2.4B model, decay using the aforementioned high-quality data unlabeled data and SFT data mixed into pre-training data, also followed by 4B token SFT.
3. B-1: 1.2B model, decay using only pre-training data, followed by 6B token SFT.
4. B-2: 1.2B model, decay using only pre-training data, followed by 12B token SFT.
5. B-3: 1.2B model, annealing using the aforementioned high-quality data + SFT data mixed into pre-training data, also followed by 6B token SFT.

The results of the experiments are shown in Table 1. We can see that, despite the A-2 and A-1 have undergone the same SFT distribution, adding SFT data to the decay stage pushes the boundary . Comparison between B-2 and B-3 demonstrate that the deficiency of only SFT is not due to the insufficient training tokens in SFT stage.

|     | C-Eval | CMMLU | MMLU | GSM8K | MATH | HumanEval | MBPP |
|-----|--------|-------|------|-------|------|-----------|------|
| A-1 | 40.0   | 41.5  | 44.6 | 27.7  | 5.1  | 27.7      | 24.4 |
| A-2 | **52.6** | **51.1** | **50.9** | **42.3** | **5.4** | **30.4** | **30.3** |
| B-1 | 40.9   | 41.5  | 47.9 | 34.2  | 7.9  | 43.9      | 30.5 |
| B-2 | 41.2   | 42.0  | 47.9 | **34.4** | 7.3 | 43.9     | 29.8 |
| B-3 | **49.1** | **46.8** | **49.6** | 31.8 | **10.5** | **44.5** | **32.8** |

Table 1: The ablation study of different training strategies.

The results indicate that the benefits of introducing high-quality data at the beginning of the decay stage are much higher than simply adding it during the SFT phase. Therefore, we recommend that specialization and enhancement of model capabilities should start from the decay phase.

# 6 Model

In this section, we begin to introduce the MiniCPM model that aggregates the aforementioned observations and techniques. Due to page limitation, the model architecture details are in Appendix F.

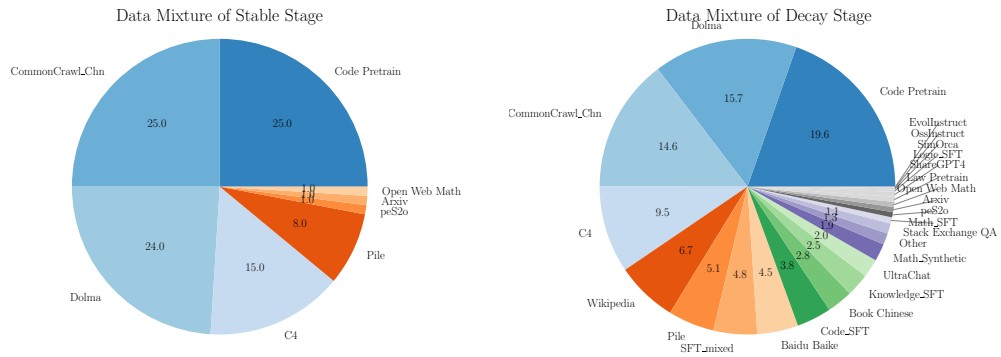

Figure 7: Data mixture of different training stages. The stable stage is shown on the left and the decay stage is shown on the right.

## 6.1 Training Stages

The overall training of the MiniCPM base model includes three stages: stable training stage, decay stage, SFT stage (Zhang et al., 2023; Wei et al., 2021). Throughout the stages, we use Adam Optimizer (Kingma & Ba, 2014).

**Stable Training Stage.** We utilize around 1T data (see Section 7 for data distribution), with the majority of the data sourced from open datasets. We use the optimal configuration discovered during the model wind tunnel experiments, WSD LRS, with a batch size of 3.93 million and a max learning rate of 0.01.

**Decay Stage.** We use a mixture of the pretraining data and high-quality SFT data. For the specific annealing form of the WSD scheduler, we employ exponential annealing, i.e. $f(s - T) = 0.5^{(s-S)/T}$, in which $T$ is set to be 5000 steps (20B tokens).

**SFT Stage.** We find it still necessary to conduct a separate SFT phase. We utilize SFT data similar to the annealing phase excluding pre-training data and train with approximately 6 billion tokens. The learning rate for SFT is aligned with the one at the end of annealing, and a WSD Scheduler with exponential decay is also employed.

## 6.2 Training Data Distribution

We introduce our training data distribution in Figure 7. In the figure, CommonCrawl_Chn in a Chinese Corpus is derived from CommonCrawl raw corpus and goes through thorough cleaning. Dolma (Soldaini et al., 2024), C4 (Raffel et al., 2019), and Pile (Gao et al., 2020; Biderman et al., 2022) are English corpora. They are deduplicated inner corpus and across corpus using MinHash algorithms (Broder, 1997). The Code Pre-train data contains the stack (Kocetkov et al., 2022) and StarCoder Li et al. (2023a), with inner deduplication and cross deduplication. In the decay stage, the data mixture contains more diverse data and proprietary data, including UltraChat (Ding et al., 2023), SlimOrca (Lian et al., 2023a;b), OssInstruct (Wei et al., 2023), EvolInstruct (Xu et al., 2023). The data with the suffix SFT is our proprietary data including LeetCode questions, Kindergarten through 12th grade (K12) textbooks and questions, etc.

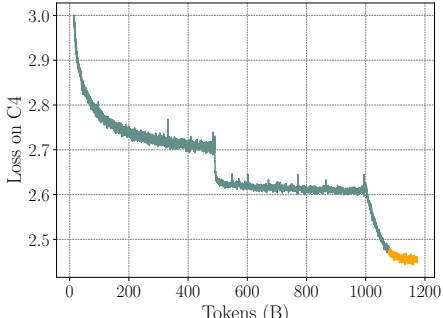 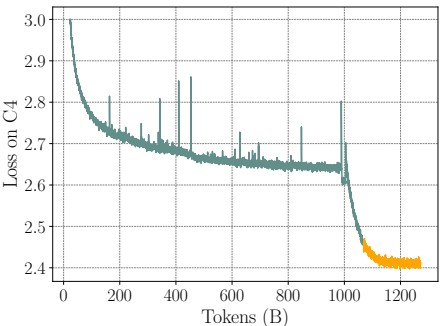

Figure 8: Loss curve on C4 dataset for MiniCPM-1.2B (Left) and MiniCPM-2.4B (Right). The orange segment at the tail of the loss curve represents the remaining decay process, which is not utilized in the released version of MiniCPM.

| Model | C-Eval | CMMLU | MMLU | HumanEval | MBPP | GSM8K | MATH |
|---|---|---|---|---|---|---|---|
| Llama2-7B | 32.42 | 31.11 | 44.32 | 12.20 | 27.17 | 13.57 | 1.80 |
| Qwen-7B | 58.96 | 60.35 | 57.65 | 17.07 | 42.15 | 41.24 | 5.34 |
| Deepseek-7B | 42.82 | 44.45 | 47.82 | 20.12 | 41.45 | 15.85 | 1.53 |
| Mistral-7B | 46.12 | 42.96 | 62.69 | 27.44 | 45.20 | 33.13 | 5.00 |
| Gemma-7B | 42.57 | 44.20 | 60.83 | 38.41 | 50.12 | 47.31 | 6.18 |
| Llama2-13B | 37.32 | 37.06 | 54.71 | 17.07 | 32.55 | 21.15 | 2.25 |
| MPT-30B | 29.34 | 32.09 | 46.56 | 21.95 | 35.36 | 10.31 | 1.56 |
| Falcon-40B | 40.29 | 41.57 | 53.53 | 24.39 | 36.53 | 22.44 | 1.92 |
| TinyLlama-1.1B | 25.02 | 24.03 | 24.3 | 6.71 | 19.91 | 2.27 | 0.74 |
| Qwen-1.8B | 49.81 | 45.32 | 43.37 | 7.93 | 17.8 | 19.26 | 2.42 |
| Qwen1.5-1.8B | **55.00** | 50.85 | 43.81 | 5.49 | 24.82 | 26.16 | 3.25 |
| Gemini Nano-3B | - | - | - | - | 27.20 | 22.80 | - |
| StableLM-Zephyr-3B | 30.34 | 30.89 | 45.90 | 35.37 | 31.85 | 52.54 | 12.12 |
| Phi-2(2B) | 23.37 | 24.18 | 52.66 | 47.56 | **55.04** | **57.16** | 3.50 |
| Gemma-2B | 29.26 | 28.56 | 38.49 | 24.39 | 29.74 | 16.83 | 3.34 |
| **MiniCPM-1.2B** | 49.14 | 46.81 | 49.63 | 44.51 | 32.75 | 31.77 | 10.60 |
| **MiniCPM-2.4B** | 51.13 | **51.07** | **53.46** | **50.00** | 47.31 | 53.83 | **10.24** |
| Model | BBH | ARC-e | ARC-c | HellaSwag | Avg | Avg$_{en}$ | Avg$_{chn}$ |
| Llama2-7B | 33.23 | 75.25[†] | 42.75 | 75.62[†] | 35.40 | 36.21 | 31.77 |
| Qwen-7B | 37.75 | 83.42 | 64.76 | 75.32[†] | 49.46 | 47.19 | 59.66 |
| Deepseek-7B | 33.38 | 74.58[†] | 42.15[†] | 75.45[†] | 39.96 | 39.15 | 43.64 |
| Mistral-7B | 41.06 | 83.92 | 70.73 | 80.43[†] | 48.97 | 49.96 | 44.54 |
| Gemma-7B | 39.19 | 89.35 | 76.79 | 79.47 | 52.22 | 54.18 | 43.39 |
| Llama2-13B | 37.92 | 78.87[†] | 58.19 | 79.23[†] | 41.48 | 42.44 | 37.19 |
| MPT-30B | 38.22 | 78.66[†] | 46.08[†] | 79.72[†] | 38.17 | 39.82 | 30.72 |
| Falcon-40B | 36.24 | 81.94[†] | 57.68 | 83.26[†] | 43.62 | 44.21 | 40.93 |
| TinyLlama-1.1B | 28.78 | 60.77[†] | 28.15[†] | 58.33[†] | 25.36 | 25.55 | 24.53 |
| Qwen-1.8B | 29.07 | 63.97[†] | 43.69 | 59.28[†] | 34.72 | 31.87 | 47.57 |
| Qwen1.5-1.8B | 28.82 | 64.86 | 45.56 | 59.39 | 37.09 | 33.57 | **52.93** |
| Gemini Nano-3B | 42.40 | - | - | - | - | - | - |
| StableLM-Zephyr-3B | 37.68 | 73.78 | 55.38 | 71.87[†] | 43.46 | 46.32 | 30.62 |
| Phi-2(2B) | **43.39** | **86.11** | **71.25** | 73.07[†] | 48.84 | **54.42** | 23.78 |
| Gemma-2B | 30.93 | 74.33 | 40.70 | 69.51 | 35.10 | 36.47 | 28.91 |
| **MiniCPM-1.2B** | 34.70 | 80.93 | 66.81 | **54.72** | 45.67 | 45.16 | 47.98 |
| **MiniCPM-2.4B** | 36.87 | 85.44 | 68.00 | 68.25 | **52.33** | 52.60 | 51.10 |

Table 2: Benchmark Score of MiniCPM-2.4B and MiniCPM-1.2B (both without RLHF). The two tables are continuous horizontally. **Avg** is over all dataset in the table, **Avg$_{chn}$** is the average of C-Eval and CMMLU while **Avg$_{en}$** is the average of remaining datasets. † means the result is tested using PPL metrics (See Appendix G. **Bold** numbers represent the best score among the SLMs. Results of Gemini Nano-3B are borrowed from Gemini et al. (2023).

## 6.3 Training Loss

The overall training loss on the C4 dataset is shown in Figure 8. We can see that as expected in the preliminary experiments, the loss decreases sharply in the decay stage. Since we use the exponential decay, the loss still drops after the learning rate drops below 10% of the max learning rate. However, since we continue to SFT the model after the decay stage, we do not utilize the final checkpoints. The checkpoints we finetune from are shown in the last checkpoint of dark green segment. The first drop in MiniCPM-1.2B is the result of enlarging batch size, which might have a similar effect as decreasing learning rate (Smith et al., 2017).

## 6.4 Evaluation

The overall evaluation utilizes our open-source tool UltraEval[3]. UltraEval is an open-source framework for assessing the capabilities of foundation models. It provides a lightweight and user-friendly evaluation system, supporting performance assessment for mainstream large models, and catering to the rapid evaluation needs of model training teams. The underlying inference and acceleration use the open-source framework vLLM (Kwon et al., 2023), and the dataset includes commonly used datasets: MMLU (Hendrycks et al., 2020) for English knowledge, CMMLU (Li et al., 2024) and C-Eval (Huang et al., 2024) for Chinese knowledge, HumanEval (Chen et al., 2021) and MBPP (Austin et al., 2021) for coding, GSM8K (Cobbe et al., 2021) and MATH (Hendrycks et al., 2021) for mathematics, and HellaSwag (Zellers et al., 2019), ARC-e (Clark et al., 2018), ARC-c (Clark et al., 2018) for commonsense reasoning, and BBH (Suzgun et al., 2022) for logic reasoning. More evaluation details can be found in Appendix G.

The overall evaluation results are in Table 7. Overall, on the mentioned datasets, we have several observations. (1) On average, MiniCPM-2.4B ranks the highest among all the SLMs. (2) MiniCPM-2.4B performs similarly to Mistral-7B-v0.1 in English but significantly outperforms Mistral-7B-v0.1 in Chinese. (3) MiniCPM-2.4B outperforms Llama2-13B except in MMLU, BBH, and HellaSwag, while MiniCPM-1.2B outperforms Llama2-7B except in HellaSwag. (4)Generally, BBH is harder for SLMs than LLMs compared to another knowledge-oriented dataset, demonstrating that reasoning ability might be more dependent on model size than knowledge. (5) Among SLMs, Phi-2 performance is on par with MiniCPM on academic-oriented datasets. This might be because their training data mostly involves textbook-style data that emphasize educational and academic scenarios. Since our pre-training data covers more distribution, we think MiniCPM is better at knowledge and ability coverage, which can be seen in Appendix K.

## 7 Conclusion

This paper introduces MiniCPM, comprising two SLMs with 2.4 B and 1.2 B non-embedding parameters, respectively. These models demonstrate superior performance compared to their larger counterparts. Our training methodologies are scalable both in terms of model and data size, offering potential applicability in the development of LLMs. The introduction of our WSD scheduler is notable for promoting continuous training, exhibiting compelling training dynamics, and enabling efficient study of scaling law. We further introduce the MiniCPM family, including DPO, long context, and MoE versions. Future directions include in-depth analysis of the loss decrease in the decay stage, and enhancing the capability of MiniCPM by scaling in both model size and data size.

---

[3]https://ultraeval.openbmb.cn/home

## Author Contributions

All authors contribute substantially to the MiniCPM project. Shengding Hu lead and participated in all aspects of the projects. This included the scaling experiments (conducted alongside Yuge Tu), babysitting the training of MiniCPM base models, and contributing to various other parts of the research. Shengding Hu wrote the paper. Chaoqun He was responsible for evaluating MiniCPM, while Ganqu Cui handled the RLHF training. Xiang Long, Zhi Zheng, Xinrong Zhang and Shengding Hu extended the context window to 128K. The MoE research was conducted by Yewei Fang and Zhi Zheng. Weilin Zhao and Kaihuo Zhang contributed to the training and inference infrastructure. The open-sourcing of MiniCPM was prepared by Yuxiang Huang and Shengding Hu. Shengding Hu, along with Chenyang Zhao, also provided analysis on the WSD scheduler's training dynamics. Zheng Leng Thai developed the tokenizer. The development of MiniCPM-V was carried out by Chongyi Wang and Yuan Yao. The training corpus of MiniCPM was prepared by Jie Zhou, Jie Cai, Shengding Hu, Zhi Zheng, and Zhongwu Zhai. The paper was proofread by Xingrong Zhang and Chaoqun He. Insightful instructions on training MiniCPM were provided by Xu Han, Ning Ding, and Zhiyuan Liu. Finally, Zhiyuan Liu, Maosong Sun, Guoyang Zeng, Chao Jia, and Dahai Li offered essential resources for the training of MiniCPM.

## Limitations

Although we have proposed a thorough study of the scaling law with SLMs, this paper does not extend to training an LLM to validate the scaling law. The application of WSD LRS on LLMs has not been fully explored to date. However, we remain optimistic about its potential advantages.

## Acknowledgement

This work is supported by the National Key R&D Program of China (No.2022ZD0160501) and the National Natural Science Foundation of China (No. 62236004, 623B2065).

MiniCPM was initially published as a technical blog on February 1st, 2024. Since then, we have received numerous insightful feedback from the community, significantly contributing to the development of this paper. We extend our gratitude to Chunting Zhou and Armen Aghajanyan for their valuable discussions. Special thanks go to Peiqin Sun and Yan Wang for their meticulous feedback on clarifying ambiguities in the blog. Additionally, we appreciate the open-source community's efforts in integrating MiniCPM into inference frameworks like llama.cpp, etc.

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

# A Additional Results in Model Wind Tunnel Experiments

## A.1 $\mu$P hyper-parameter search

We conduct an extensive Bayesian search over a set of predefined parametric spaces. For efficiency, we search for the $N = 0.009B$ model. In our pilot experiments, we confirm that when hyper-parameter optimization is conducted using datasets scaled at magnitudes of 10N and 20N, there is a consistency observed in the efficacy of hyper-parameters. Therefore, we train the models with $|D| = 10N = 0.09B$ tokens. Meanwhile, we also try QK-Norm (Henry et al., 2020) and independent weight decay (Loshchilov & Hutter, 2017) as well to stabilize the learning rate. The overall results are shown in Figure 9. After applying the QK-norm, we observe a significant decrease in the learning rate sensitivity similar to Wortsman et al. (2023). However, as the MiniCPM project itself is an SLM, we do not require low learning rate sensitivity as long as we find the best learning rate with TensorProgram (Yang et al., 2022; 2023). Therefore, we do not introduce QK-norm and independent weight decay in later experiments of MiniCPM. In Figure 9, we identify the best hyper-parameters for $scale\_depth = 1.4$, $scale\_emb = 12$, $init\_std = 0.1$, and $lr = 0.01$.

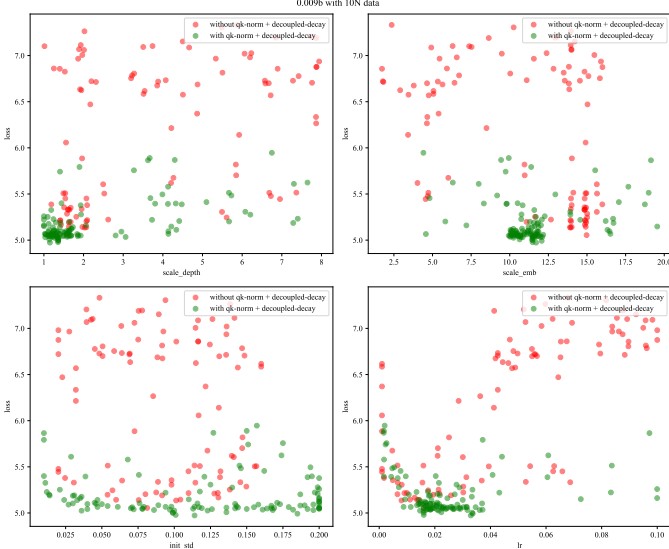

Figure 9: Grid search over the $\mu$P parameterization spaces.

| Name | Specific Operation |
|---|---|
| Embedding Output Scaling | Multiply the output of the embedding by $scale\_emb$ |
| Residual Connection Scaling | Scale the output tensor of a block before adding to each residual connection in each layer by $scale\_depth / \sqrt{\text{num\_layers}}$ |
| Initialization of Tensors | Set the initialization standard deviation of each two-dimensional tensor parameter to $init\_std / \sqrt{d_m / d_{base}}$, and set other parameters' initialization to 0.1 |
| Learning Rate Scaling of Tensors | Adjust the learning rate of each two-dimensional tensor parameter to $1/(d_m / d_{base})$ times the learning rate of other parts (or the overall learning rate) |
| LM Head Scaling | Adjust the output logits to $1/(d_m / d_{base})$ times the original value |

Table 3: List of operations used when applying tensor program techniques.

## A.2 Comment on Optimal Batchsize

In Kaplan et al. (2020), OpenAI studies the relationship between the loss function and the number of tokens. In their experiments, they assume that consuming more steps is equivalent to consuming more time. Under this assumption, OpenAI defines a critical batch size that achieves a certain loss without consuming too many steps or tokens. This rationale is valid if the experiments are provided with unlimited GPUs (at least within the scope of the experiments). Since GPUs are unlimited, enlarging batch size will not increase the single-step duration but will decrease the total number of steps. However, in our experiment, since we have a fixed resource (number of GPUs), we observe that doubling the batch size almost equals doubling the single-step time. Therefore, enlarging batch size to decrease total training steps has minimal effect on the total training time. In light of this observation, we drop the goal of "not consuming too many steps" and turn towards minimizing the token quantity to achieve the lowest loss, instead.

The observation regarding the estimation of optimal batch size in relation to loss resembles the "Chicken-and-egg" paradox. Practically, there's often a preliminary estimate of the achievable loss for a given model size, informed by prior knowledge of preliminary experiments. However, there is potential for the development of more refined estimation procedures in the future.

The optimal batch size and optimal learning rate are likely to be not independent. To overcome this correlation, we do a preliminary study on the learning rate first, then choose an optimal learning rate to do a batch size experiment, and use batch size scaling to do the learning rate again. This is a bit like the Coordinate Descent optimization method. However, more rigorous methods are welcomed in future work.

## A.3 Model Architecture in Model Wind Tunnel Experiments

We list the model configuration used in the model wind tunnel experiments in Table 4. The "shape" of the model, i.e., model width compared to model depth is kept as similar as possible to avoid any potential performance variation.

| Name | N (B) | $d_m$ | $d_{ff}$ | $d_h$ | $n_h$ | $L$ |
|------|-------|-------|----------|-------|-------|-----|
| 9M | 0.009 | 320 | 800 | 64 | 5 | 8 |
| 30M | 0.036 | 512 | 1280 | 64 | 8 | 12 |
| 70M | 0.066 | 640 | 1600 | 64 | 10 | 14 |
| 0.1B | 0.109 | 768 | 1920 | 64 | 12 | 16 |
| 0.17B | 0.166 | 896 | 2240 | 64 | 14 | 18 |
| 0.2B | 0.241 | 1024 | 2560 | 64 | 16 | 20 |
| 0.5B | 0.499 | 1344 | 3360 | 64 | 21 | 24 |

Table 4: Model configurations and training configurations of the models in the scaling curve. N(B) represents the number of non-embedding parameters of the models, measured in billions.

# B Additional Illustration on WSD LRS

## B.1 Learning Rate Paradigm for Different LRSs

In this paper, we describe three kinds of LRSs, $Cosine(T)$, $CosineLoop(T)$, and $WSD(T, D)$. Cosine and Cosine Loop take the form of the following:

An illustrative learning rate diagram for WSD and Cosine Scheduler is shown in Figure 10.

$$Cosine(T;s) =$$
$$\begin{cases} \frac{s}{W}\eta, & s < W \\ 0.9\eta cos(\pi\frac{s}{T}) + 0.1\eta, & W < s < T \\ 0.1\eta, & s > T \end{cases}$$

$$CosineLoop(T;s) =$$
$$\begin{cases} \frac{s}{W}\eta, & s < W \\ 0.9\eta cos(\pi\frac{s}{T}) + 0.1\eta, & W < s \end{cases}$$

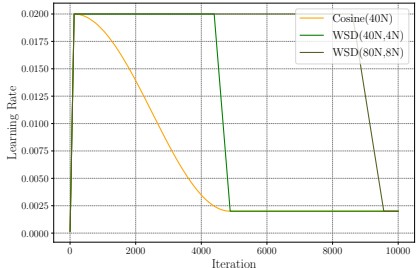

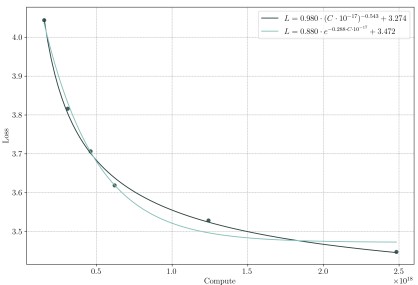

Figure 10: Illustrative comparison between Cosine LRS and WSD LRS. The WSD LRS with different end steps share the same stable training stage.

Figure 11: We use two different function forms to fit the data scaling law achieved by WSD LRS and choose power law as the best fit.

## B.2 Fitting the Data Scaling Law

In this section, we describe the fitted data scaling law for continue training with WSD LRS. Each point in Figure 11 is the end of the decay stage in WSD LRS with a different end step. We try two function forms: exponential and polynomial. The fitted result shows that the polynomial scaling law is still the best for continue training.

## B.3 Individual Figure for Model-Data Scaling Law

For each task and model, the scaling law $L(N, D)$'s fitness with real loss values along the data axis is plotted in Figure 12.

## B.4 Analysis of Llama2's Data-to-Model Ratio

As mentioned in Section E, we analyze Llama2's Data-to-Model Ratio based on their training loss curve. The extracted loss is plotted on the left of Figure 13. We convert the x-axis to computation Flops to compare the computed optimal regime on the right part of the Figure.

## C MiniCPM's Vocabulary

Despite being small in parameter size, MiniCPM targets modeling diverse data distribution, excelling in English and Chinese. Therefore, our vocabulary is relatively large. For the 2.4B model, we use a tokenizer consisting of 122,753 tokens (denoted by MiniCPMTokenizer-120K). This vocabulary is constructed from extensive and diverse language data, utilizing the sentencepiece library [4] for Byte Pair Encoding (BPE) (Sennrich et al., 2016), and includes special symbols like traditional Chinese characters, rare characters, emojis, and special symbols such as Greek letters, Cyrillic letters, etc.

For the SLM, the embedding parameters will take up a lot of parameter space if the vocabulary is large. Therefore, for our 1.2B model, we use a smaller vocab MiniCPMTokenizer-70K. Compared to the MiniCPMTokenizer-120K tokenizer, we have re-trained the tokenization

---

[4]https://github.com/google/sentencepiece

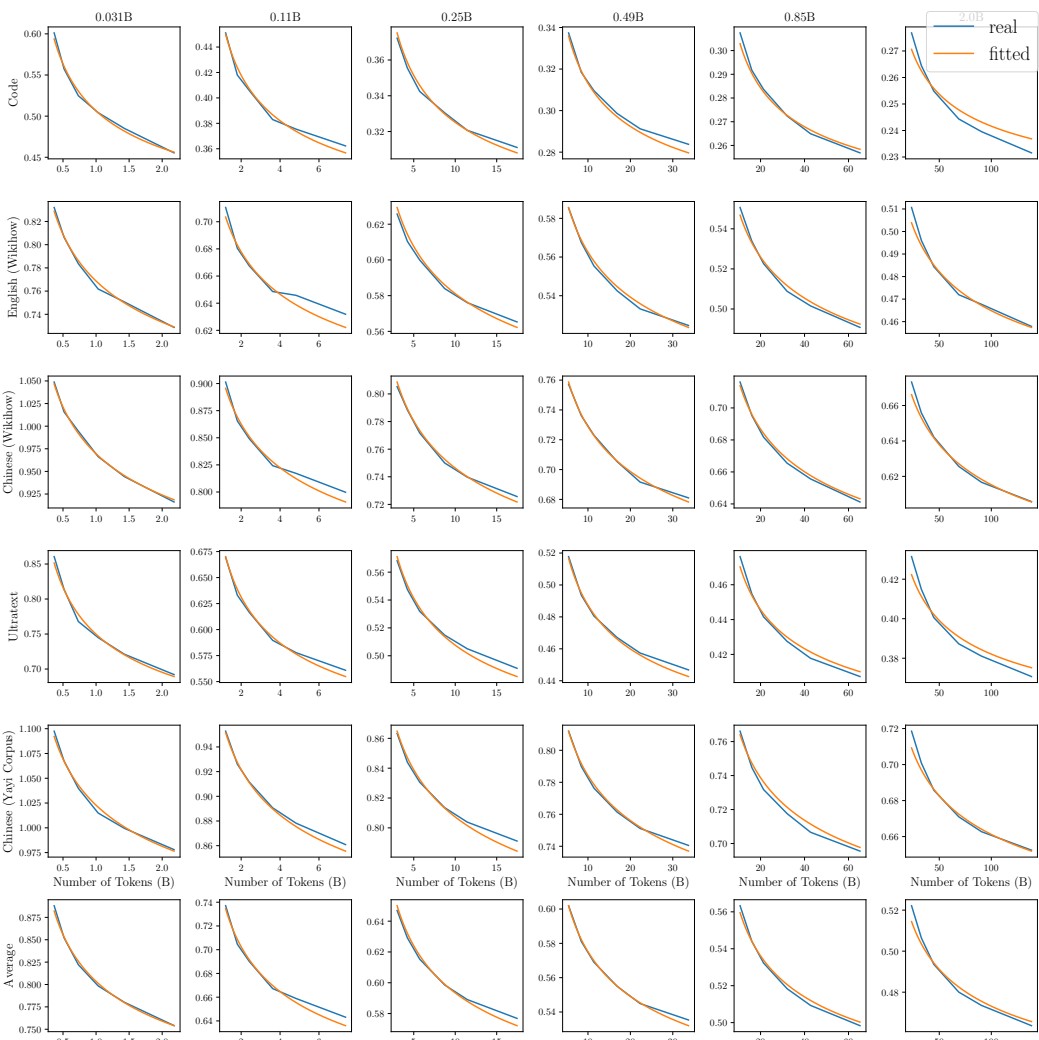

Figure 12: The fitted scaling law plotted along the data amount axis for each model and each task. The fitted result is satisfying except for the last checkpoints of the 0.11B and 0.25B model.

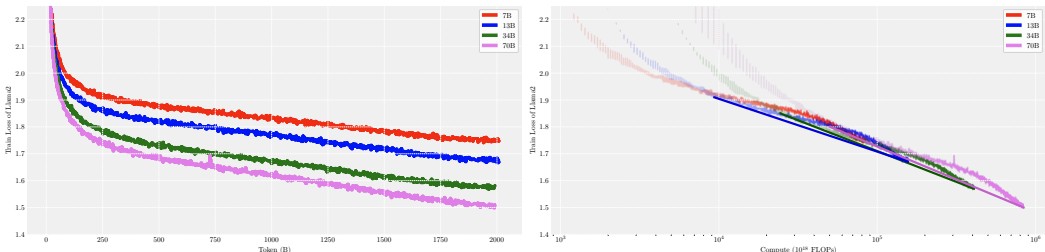

Figure 13: We extract the training loss data from Llama2 paper (left part) and estimate the compute optimal $\frac{D_{opt}}{N_{opt}}$ in their paper using the right part. The straight lines are plotted to estimate the optimal loss envelope assuming using WSD Scheduler.

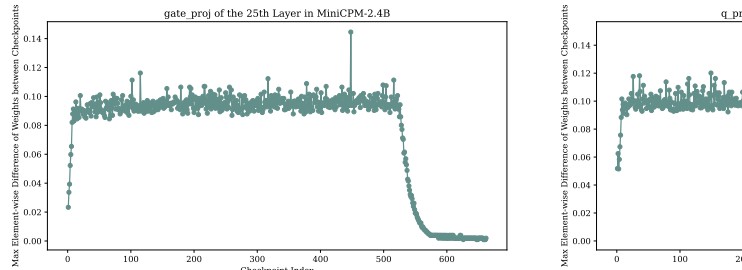

Figure 14: Max Difference of Checkpoints.

on the same documents, while setting the max number of vocabs to 64,000. For the special characters, we only add the traditional Chinese characters, emojis, and special symbols, but leave out the rare characters in Chinese.

We conduct evaluations on 300,000 documents in Chinese, English, code, and academic papers that are not in the training set of the Tokenizer. The MiniCPM-120K tokenizer achieves the highest compression ratio (Bytes/Tokens).

|  | Baichuan2 | ChatGLM2 | Llama2 | MiniCPM-120K | MiniCPM-70K |
|---|---|---|---|---|---|
| Vocab Size | 125,696 | 64,794 | 32,000 | 122,753 | 73,440 |
| **Compression Rate** (Bytes/Tokens) | | | | | |
| Chinese | 3.64 | 3.54 | 1.87 | **3.73** | 3.56 |
| English | 4.12 | 4.02 | 3.78 | **4.14** | 4.02 |
| Code | 2.71 | 2.71 | 2.74 | **2.81** | 2.76 |
| Paper | 2.74 | 2.88 | **2.97** | 2.93 | 2.88 |
| Average | 3.30 | 3.29 | 2.84 | **3.40** | 3.31 |

Table 5: Compression ratio comparison.

# D   Analysis of the Decay Stage

In this section, we provide a brief analysis of the loss drop in the decay stage, examining it through the prisms of checkpoint updates and gradient information. We calculate the maximum weight element update $max_{ij}(W_{ij}^{(t+1)} - W_{ij}^{(t)})$ across all weight matrices in the MiniCPM-2.4B (Introduced in Section F). As depicted in Figure 14, the updates exhibit a robust correlation with the learning rate's magnitude. Notwithstanding the illustration of the two submodules (gate_proj and q_proj module of the 25th layer), this pattern is prevalent across every layer and submodule within the network. This observation may not be trivial: the model checkpoints experience significant updates preceding the learning rate's decay, yet the loss exhibits minimal reduction. Conversely, during the decay stage, despite less pronounced weight alterations, there is an accelerated decrease in loss.

Further examination of the gradient data is undertaken by training a 0.2B model, meticulously recording every step gradient information, and evaluating the differences between consecutive steps, thereby providing an approximation of second-order gradient information. We treat the gradient at step $t$ as a flattened vector $\mathbf{g}^{(t)}$, and the parameter (also flattened as a vector $\mathbf{x}^{(t)}$ ) update between step $t$ and $t + 1$ is $\mathbf{v}^{(t)} = \mathbf{x}^{(t+1)} - \mathbf{x}^{(t)}$. The gradient norm take the $L2$ norm of the gradient $\|\mathbf{g}^{(t)}\|$, gradient inner product is $\mathbf{g}^{(t+1)} \cdot \mathbf{g}^{(t)}$, the cosine of the gradient's angle is given by $\frac{\mathbf{g}^{(t+1)} \cdot \mathbf{g}^{(t)}}{\|\mathbf{g}^{(t+1)}\|\|\mathbf{g}^{(t)}\|}$. Imaging the optimization process as a trajectory over a high-dimension manifold, first order directional derivative along the

trajectory is computed as $D_1 = \frac{\mathbf{g}^{(t+1)} \cdot \mathbf{v}^{(t)}}{\|\mathbf{v}^{(t)}\|}$, and the second order directional derivative is $D_2 = \frac{(\mathbf{g}^{(t+1)} - \mathbf{g}^{(t)}) \cdot \mathbf{v}^{(t)}}{\|\mathbf{v}^{(t)}\|^2}$. $D_1, D_2$ enables an approximate estimation of the loss curvature on the trajectory, $K = \frac{|D_2|}{(1+D_1^2)^{\frac{3}{2}}}$. The results of these statistics over time are shown in Figure 15. We can see that the gradient norm diminishes during the decay phase, and upon commencement of this stage, the cosine between gradients predominantly assumes positive values, suggesting that in the decay phase, model parameters undergo consistent changes across steps. Concerning directional derivatives, it is remarkable that the first-order directional derivative diminishes exponentially with each step, aligning closely with the learning rate, while the second-order directional derivative exhibits a slight increase in magnitude. The curvature of the loss function also increases by a magnitude, indicating the proximity to a local optimum. These findings potentially offer a deeper insight into the shape of optimization space, a subject reserved for future exploration.

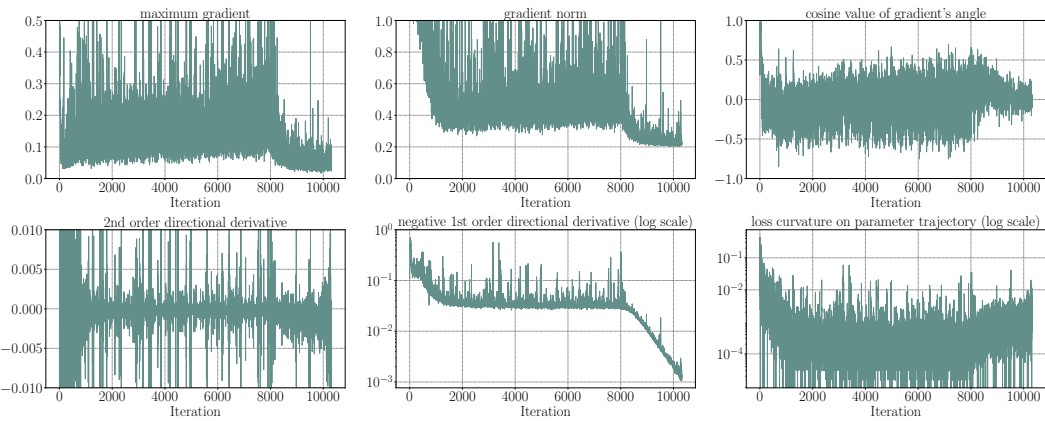

Figure 15: Gradient statistics over the training of a 0.2B model using WSD LRS. The exponential decay stage begins at 8000 steps.

# E   Measuring the Scaling Law with WSD LRS

Scaling laws serve as a fundamental guiding principle in the development of LLMs. Although these scaling laws exhibit variability in specific coefficients due to diverse configurations across model series, the compute optimal data-to-model ratio remains a meaningful metric across different scaling law functions, which "marginalizes" out the specific value of loss. Regarding this ratio, Kaplan et al. (2020) posit that a tenfold increase in model scale should equate to a singlefold increase in data scale. Conversely, Hoffmann et al. (2022) argue for the same scaling rate between model size and data size. What's more, current models such as LLama 2 (Touvron et al., 2023), train much more data than what Hoffmann et al. (2022) claims, still yielding considerable performance gain. Indicating a higher data-to-model ratio.

This unaddressed uncertainty stems from the challenges inherent in training multiple models of varying sizes and data sizes in traditional scaling experiments. Previously, if the average cost of training one model size on one data size is $C$, then conducting the scaling experiments with $m$ model sizes and $m$ data sizes takes approximately $O(m^2)C$.

In this section, we introduce the utilization of the WSD scheduler as an effective approach to explore the scaling law with linear cost ($O(mC)$). Since the WSD scheduler has the advantage of arriving at the optimal loss of Cosine LRS after decaying from stable stage checkpoints of any step, we are now able to precisely measure the optimal scaling properties without re-training the models from scratch to different amounts of tokens, thus making the scaling law measurement much more efficient along the data axis.

We measure the scaling law along the data and model axes by training SLMs of 6 sizes ranging from 0.04B to 2B, each with 6 decayed models starting from the checkpoint of $10N$ to $60N$ data during the stable training stage ($N$ is the respective model size). The final loss is evaluated on five held-out evaluation datasets. To potentially compare the loss when the model uses different tokenizers, we take the average of loss by a number of bytes instead of a number of tokens, following Achiam et al. (2023). The final loss of each pair of data size and model size is shown in the blue lines in Figure 12.

Then we fit the losses with model size $N$ and data size $D$ following Hoffmann et al. (2022) using scipy `curvefit` function:

$$L(N, D) = C_N N^{-\alpha} + C_D D^{-\beta} + L_0 \qquad (2)$$

The fitted curve along the data axis for each dataset and each checkpoint are shown in orange lines in Figure 12. Then we have the optimal model size $N_{opt}$, dataset size $D_{opt}$, given a fixed amount of compute $C = 6ND$ (Rae et al., 2021) as:

$$\frac{N_{opt}}{D_{opt}} = K^2 \left(\frac{C}{6}\right)^\eta, \qquad (3)$$

where $K = \left(\frac{\alpha C_N}{\beta C_D}\right)^{\frac{1}{\alpha+\beta}}$, and $\eta = \frac{\beta-\alpha}{\alpha+\beta}$. The derivation of $N_{opt}$ closely follows Hoffmann et al. (2022) by substituting $D$ with $\frac{C}{6N}$ in Equation 2, and minimize $L(N)$ given $C$. A similar way is adopted for $D_{opt}$. From Equation 3, when $\alpha = \beta$, $N_{opt}/D_{opt}$ is a constant, supporting Hoffmann et al. (2022)'s claim, and when $\alpha < \beta$, we should emphasize more on parameter scaling (Kaplan et al., 2020), and vise versa.

In our experiments, the fitted relationship between loss and $N, D$ is shown in the contour plot of equal loss in Figure 17. The equation of fitted scaling law is shown in the first text box in each subplot. We can see that in all evaluation corpora, we have $\beta < \alpha$. More specifically, on average, we have $\alpha = 0.29, \beta = 0.23, K^2 = 0.01, \eta = -0.10$ (Note that N is under $10^9$, D is under $10^9$, and C is under $10^{18}$). Since $\alpha$ is slightly larger than $\beta$, this result shows that as the computation scale, we should slightly emphasize more on data scaling than model scaling, which aligns with Hoffmann et al. (2022).

As for the concrete data-to-model ratio $\frac{D_{opt}}{N_{opt}}$, we notice that there is a huge gap in compute optimal regime between ours and Hoffmann et al. (2022) despite that the trend of $\frac{D_{opt}}{N_{opt}}$ with compute $C$ is aligned between ours and theirs. Specifically, the data size should be 192 times larger than the model size on average, as opposed to 20 times in Hoffmann et al. (2022). We note that this aligns with the observation in Section 4.3 and Figure 6.

With respect to the large deviation from Chinchilla Optimal $\frac{N_{opt}}{D_{opt}}$, we notice that their scaling experiment was conducted in a not very recent configuration. To compare with more recent configuration such as Llama2 (Touvron et al., 2023), we extract the training loss data from Llama2 paper (left part) in Appendix Figure 13 and estimate the compute optimal $\frac{D_{opt}}{N_{opt}}$ in their paper using the right part of Figure 13. Since they use Cosine LRS, the loss is not optimal in the middle of the training, depicted by the concave curve during training in the right figure of Figure 13. We fill the concave part with a straight line to estimate the optimal loss envelope if they had used the WSD LRS. After that, the compute model size should roughly be the regime in which a model's loss curve is about to intersect with a larger model's loss curve. With this intuition, the 13B model is about to intersect with the 34B model at $10^5$ EFlops ($10^{18}$ Flops), and the 34B model is about to intersect with the 70B model at $5 \times 10^5$ EFlops. Therefore, we estimate the $\frac{D_{opt}}{N_{opt}}$ to be roughly $\frac{5\times10^5}{6\times34^2} \sim \frac{10^5}{6\times13^2}$, which is $70 \sim 100$. Therefore, under this approximate comparison, their data-model ratio is closer to ours. And our configuration can absorb more data into a smaller model compared to previous ones. However, we note that the above estimates are only a rough one.

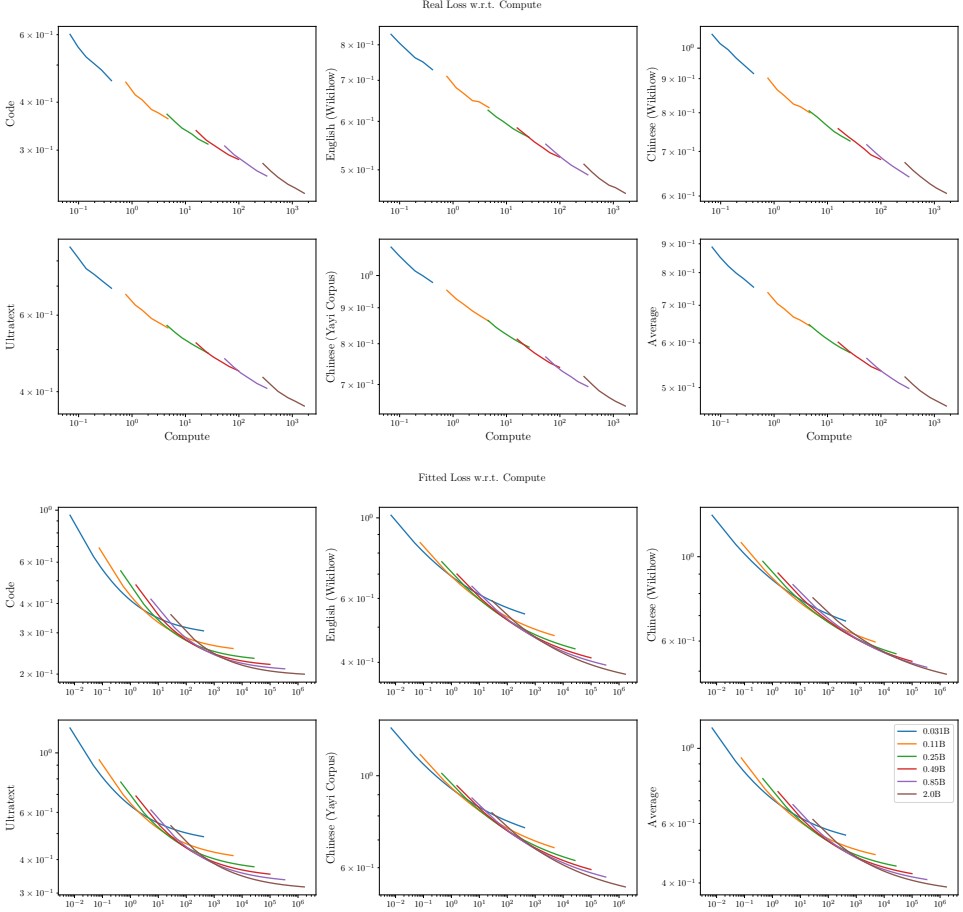

Figure 16: The result of scaling experiments with WSD Scheduler (above) and the fitted scaling curve (below). The x-axis is the computation Flops $C = 6ND$, each color of the line represents the same model with different computation Flops. We can see that smaller models are better than larger models when the Flops are small and worse when the Flops are large. Thus models of different sizes will intersect with each other in the plot around the compute optimal regime.

A larger data-to-model ratio means that we can absorb more data into a smaller model than we previously thought, which is more efficient for inference and deployment. We hope WSD LRS will help more researchers explore $L(N, D)$ with less effort and make the relationship clearer in LLMs.

## F  Model Details

**Vocabulary.**  We use two tokenizers of 122,753 vocabulary size for MiniCPM-2.4B and 73,440 vocabulary for MiniCPM-1.2B. A small vocabulary for 1.2B favors efficiency without harming much performance. Details of the tokenizers are in Appendix C. Including the embedding parameters increases total parameters by 0.3B and 0.2B respectively.

**Shared Input-output Layer.**  For SLM, the embedding takes up a large parameter space. To make the model parameters smaller, we use the Embedding Sharing techniques for both MiniCPM-2.4B and MiniCPM-1.2B.

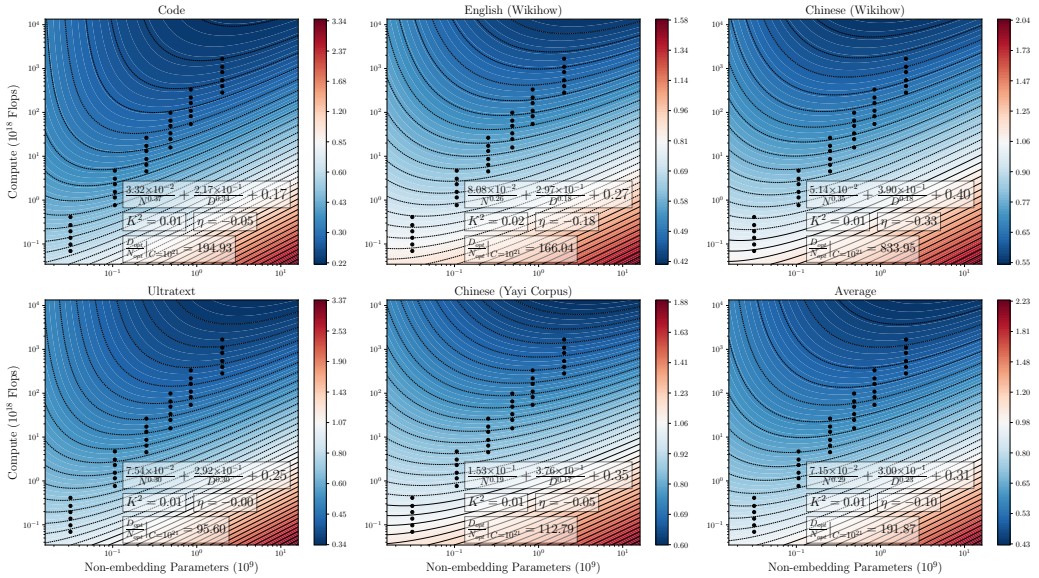

Figure 17: The fit result of the scaling experiment with WSD Scheduler. The black dots in a horizontal line denote the decayed checkpoints in different compute within the same model size. The *C* in the legend is expressed in Flops units for semantic completeness and does not use EFlops units. The C in Y-axis is in EFlops.

**Deep-and-thin Network.** We train MiniCPM-2.4B before training MiniCPM-1.2B. When training MiniCPM-2.4B, we adopt a deeper and thinner architecture compared to Phi-2 (Javaheripi & Bubeck, 2023) (40 layers compared to 32 layers). Recently, Liu et al. (2024) propose to train deep and thin networks for SLMs, which aligns with our perspective. Therefore, we further make the architecture deeper and thinner for MiniCPM-1.2B.

**Group Query Attention.** We train MiniCPM-2.4B without modification on the attention layer. Whereas we apply Group Query Attention (Ainslie et al., 2023) to MiniCPM-1.2B, inspired by Liu et al. (2024), to further reduce the parameters number.

| Model | N (B) | $d_m$ | $d_{ff}$ | $d_h$ | $n_q$ | $n_{kv}$ | L | Batch size (M) | Tokens (T) |
|---|---|---|---|---|---|---|---|---|---|
| MiniCPM-1.2B | 1,247,442,432 | 1,536 | 3,840 | 64 | 24 | 8 | 52 | 2M → 4M | 1.1T |
| MiniCPM-2.4B | 2,442,057,984 | 2,304 | 5,760 | 64 | 36 | 36 | 40 | 4M | 1.1T |

Table 6: Model configurations for MiniCPM. N (B), $d_m$, $d_{ff}$, $d_h$, $n_q$, $n_{kv}$, L, Batch size (M), Tokens (T) represents the number of non-embedding parameters of the model, model hidden dimension, feedforward layer bottleneck dimension, attention head dimension, number of queries, number key/values, number of layers, training batch size, total training tokens.

## G Evaluation Details

UltraEval is an open-source framework for assessing the capabilities of foundation models. It provides a lightweight and user-friendly evaluation system, supporting performance assessment for mainstream large models, and catering to the rapid evaluation needs of model training teams. The underlying inference and acceleration use the open-source framework vLLM (Kwon et al., 2023), and the dataset includes commonly used datasets: MMLU (Hendrycks et al., 2020) for English knowledge, CMMLU (Li et al., 2024) and C-Eval (Huang et al., 2024) for Chinese knowledge, HumanEval (Chen et al., 2021) and MBPP (Austin et al., 2021) for coding, GSM8K (Cobbe et al., 2021) and MATH (Hendrycks et al., 2021) for mathematics, and HellaSwag (Zellers et al., 2019), ARC-e (Clark et al., 2018),

ARC-c (Clark et al., 2018) for commonsense reasoning, and BBH (Suzgun et al., 2022) for logic reasoning.

Due to the difficulty of standardizing evaluations for large models and the lack of publicly available prompts and test codes for many models' evaluations, we try our best to adapt the evaluation methods to suit various model types. Specifically, we start from a standardized input prompt during testing and adjust it according to each model's appropriate input-output template. The **evaluation scripts and prompts are also open-source** in our repository, and we welcome developers to continually improve our evaluation methods.

When testing QA tasks (ARC-e, ARC-c, HellaSwag), two approaches are typically employed. The first involves using Perplexity (PPL): we extend each option as the continuation of the question and use the PPL of the option as the selection criterion. The second is direct generation, where the model directly outputs answer options. We observe significant differences in results obtained using these two methods. MiniCPM performs similarly in direct generation and PPL tests, with better performance in direct generation. On the other hand, Mistral-7B-v0.1 performs better in PPL tests but exhibits poorer performance in direct generation. To address this phenomenon, when reporting the scores for each model, we adopt the score from the evaluation method that yields the highest score, ensuring fairness in comparison.

## H   Quantization

We conduct 4-bit quantization on our model. We do not quantize the parameters of embedding and layer normalization, since the performance of the model is sensitive to these parameters. Therefore, we only need to quantify each weight matrix. Denote the weight matrix as $\mathbf{W} \in \mathbb{R}^{d_{out} \times d_{in}}$. We group every $G$ consecutive parameter at the $d_{in}$ dimension and form $d_{in}/G$ group. Then we quantize each group of the parameters separately. For each group parameter $\mathbf{w}$, we calculate the quantization scale and zero point as follows:

$$\text{scale} = \frac{\max(\mathbf{w}) - \min(\mathbf{w})}{2^4 - 1}, \text{zero} = -\frac{\min(\mathbf{w})}{scale} - 2^3.$$

Group parameter $\mathbf{w}$ are then quantized to

$$\hat{w} = quant(w) = round(\frac{w}{\text{scale}} + \text{zero}),$$

where *round* operation round a floating point to nearest integer. The dequantization operation is approximately the reverse of the quantization method, which is

$$dequant(\hat{w}) = \text{scale}(\hat{w} - \text{zero}).$$

Finally, matrix $\mathbf{W} \in \mathbb{R}^{d_{out} \times d_{in}}$ is quantized to int4 $\hat{\mathbf{W}} \in \mathbb{R}^{d_{out} \times d_{in}}$, float **scale** $\in \mathbb{R}^{d_{out} \times \frac{d_{in}}{G}}$ and float **zero** $\in \mathbb{R}^{d_{out} \times \frac{d_{in}}{G}}$.

To reduce the quantization loss, we adopt GPTQ (Frantar et al., 2022) to apply weight calibration. We sample calibration data $\mathbf{X}$ from a similar distribution of SFT data. The quantization objective is to minimize the disturbance of the quantization $\|\mathbf{W}\mathbf{X} - dequant(\hat{\mathbf{W}})\mathbf{X}\|_2^2$. We follow GPTQ to quantize weight iteratively and update the remaining non-quantized weight by

$$\delta_F = -\frac{w_q - dequant(quant(w_q))}{[H_F^{-1}]_{qq}} \cdot (H_F^{-1})_{:,q},$$

where $q$ is the quantization position in the current iteration while $F$ denotes the remaining non-quantized weights. $H_F$ is the hessian matrix of the objective function.

# I  MiniCPM Family

In this section, we introduce the other models that build on MiniCPM base models. Specifically, we trained the aligned model, long-context model, and MoE model for MiniCPM 2.4B.

## I.1  MiniCPM-DPO

After SFT, we employ DPO (Rafailov et al., 2024) for human preference alignment of the model. During this stage, UltraFeedback (Cui et al., 2023) is utilized as the primary alignment dataset, and a proprietary preference dataset is constructed to enhance the model's code and mathematical capabilities. We conduct one epoch of DPO training with a learning rate of $1 \times 10^{-5}$ and utilize a Cosine LRS since we have a pre-defined training step.

After applying DPO for preference alignment, the model's score on MTBench (Zheng et al., 2024) increased from 6.89 after SFT to 7.25, surpassing even large models such as Llama2-70B-Chat (see Figure 18). However, we also noticed that the performance on benchmarks is slightly compromised, which is known as the alignment tax (Askell et al., 2021).

| Model | C-Eval | CMMLU | MMLU | HumanEval | MBPP | GSM8K | MATH |
|---|---|---|---|---|---|---|---|
| ChatGLM2-6B | 52.05 | 49.21 | 45.77 | 10.37 | 9.38 | 22.74 | 5.96 |
| Mistral-7B-Instruct-v0.1 | 38.06 | 36.96 | 53.56 | 29.27 | 39.34 | 28.73 | 3.48 |
| Mistral-7B-Instruct-v0.2 | 42.55 | 41.92 | 60.51 | 36.59 | 48.95 | 40.49 | 4.95 |
| Qwen-7B-Chat | 58.57 | 57.23 | 56.03 | 15.85 | 40.52 | 42.23 | 8.3 |
| Yi-6B-Chat | 70.88 | 71.11 | 62.95 | 14.02 | 28.34 | 36.54 | 3.88 |
| Baichuan2-7B-Chat | 53.28 | 53.50 | 53.00 | 21.34 | 32.32 | 25.25 | 6.32 |
| Deepseek-7B-chat | 46.95 | 49.72 | 51.67 | 40.85 | 48.48 | 48.52 | 4.26 |
| Llama2-7B-Chat | 34.54 | 32.64 | 47.64 | 14.02 | 27.40 | 21.15 | 2.08 |
| **MiniCPM-2.4B-DPO** | 48.64 | 48.37 | 53.05 | 51.22 | 48.01 | 53.37 | 9.86 |

| Model | BBH | ARC-e | ARC-c | HellaSwag | Avg | Avg$_{en}$ | Avg$_{chn}$ |
|---|---|---|---|---|---|---|---|
| ChatGLM2-6B | 32.60 | 74.45 | 56.82 | 58.48[†] | 37.98 | 35.17 | 50.63 |
| Mistral-7B-Instruct-v0.1 | 39.52 | 81.61 | 63.99 | 73.47[†] | 44.36 | 45.89 | 37.51 |
| Mistral-7B-Instruct-v0.2 | 39.81 | 86.28 | 73.38 | 84.55[†] | 50.91 | 52.83 | 42.24 |
| Qwen-7B-Chat | 37.34 | 64.44[†] | 39.25[†] | 74.52[†] | 44.93 | 42.05 | 57.90 |
| Yi-6B-Chat | 37.43 | 84.89 | 70.39 | 74.60[†] | 50.46 | 45.89 | 71.00 |
| Baichuan2-7B-Chat | 37.46 | 79.63 | 60.15 | 69.23[†] | 44.68 | 42.74 | 53.39 |
| Deepseek-7B-chat | 35.70 | 76.85 | 63.05 | 76.68[†] | 49.34 | 49.56 | 48.34 |
| Llama2-7B-Chat | 35.54 | 74.28 | 54.78 | 75.65[†] | 38.16 | 39.17 | 33.59 |
| **MiniCPM-2.4B-DPO** | 36.22 | 85.02 | 68.17 | 65.67 | 51.60 | 52.29 | 48.51 |

Table 7: Benchmark scores for MiniCPM-2.4B-DPO compared with larger chat models.

## I.2  MiniCPM-128K

Tasks involving lengthy contexts depend on the implicit information within these contexts, circumventing the need for the extensive knowledge often absent in SLMs. In this section, we expand the context length of MiniCPM-2.4B from 4,096 to 128,000 tokens, illustrating the capability of SLM to effectively process long contexts.

**Initialization.** For the initialization, we disable sharing embeddings between input and output, primarily to accommodate vocabulary parallelism essential for training with long context. The LM head is initialized from the input embedding.

**Training.** Similar to MiniCPM, MiniCPM-2.4B-128K utilizes the WSD as its learning rate scheduler and reuses the last checkpoint of the stable training stage of MiniCPM-2.4B. Concerning training data, we categorize the dataset distribution detailed in Section 6.2 into "short data" and "long data". We classify books, wikis, and papers as "long data", and the other as the "short data". The training comprises 44% long data and 56% short

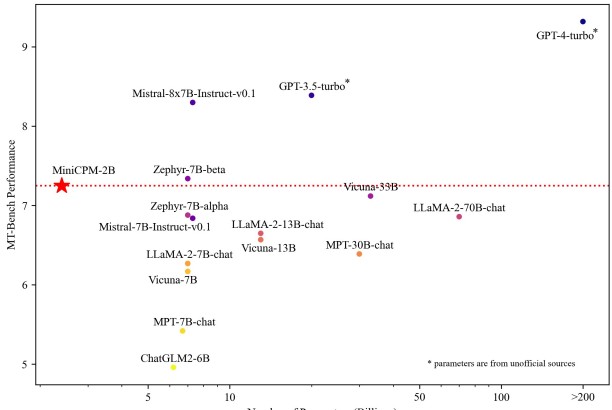

Figure 18: MTBench score of MiniCPM-DPO-2.4B surpasses many models of larger size.

data for continued training. For the extension of long contexts, we apply Adjusted Base Frequency (ABF) (Xiong et al., 2023) in the 4K to 32k range and employ NTK-Aware RoPE Scaling (bloc97, 2023) and curriculum learning from 32K to 128K. Both two stages involve future training. Furthermore, as indicated in Yi Tech Report (Young et al., 2024) and Zebra (Song et al., 2023), we use of synthetic long QA data that significantly enhances model performance in context-aware tasks.

**Evaluation.** We evaluate MiniCPM-2.4B-128K in ∞Bench (Zhang et al., 2024b), a pioneering benchmark for long context evaluations. The tasks in ∞Bench (Zhang et al., 2024b) extend beyond typical retrieval tasks and challenge the model with long context reasoning. We can see in Table 8, we achieve comparable results in Mistral-7B-Instruct-v0.2 (ABF1000w) and outperform ChatGLM3-6B-128K despite being 2.5 times smaller.

| Model | Passkey | Number String | KV Re- trieval | Long Book Choice Eng | Long Book QA Chn | Long Book QA Eng | Long Book Sum Eng |
|---|---|---|---|---|---|---|---|
| LWM-Text-128K | 100 | 97.8 | 0.6 | 28.82 | 15.93 | 14.31 | 9.99 |
| Yarn-Mistral-7b-128K | 92.71 | 56.61 | 0 | 27.95 | 15.49 | 9.55 | 9.06 |
| Mistral-7B-Instruct-v0.2(ABF 1000w) | 100 | 78.98 | 3.6 | 37.12 | 11.74 | 17.37 | 21.12 |
| Yi-6B-200K | 100 | 94.92 | 0 | 36.68 | 15.07 | 9.2 | 0.92 |
| ChatGLM3-6B-128K | 89.93 | 99.66 | 5.2 | 46.29 | 10.7 | 8.38 | 25.91 |
| **MiniCPM-2.4B-128K** | 98.31 | 99.83 | 9 | 29.69 | 23.06 | 16.33 | 15.73 |

| Model | Long Di- alogue QA Eng | Math Calc | Math Find | Code Debug | Code Run | Avg | Avg w/o Code & Math |
|---|---|---|---|---|---|---|---|
| LWM-Text-128k | 1.5 | 0 | 3.43 | 20.05 | 1 | 24.45 | 33.62 |
| Yarn-Mistral-7b-128k | 7.5 | 0 | 17.14 | 0.76 | 1.25 | 19.84 | 27.36 |
| Mistral-7B-Instruct-v0.2(ABF 1000w) | 9.5 | 0 | 29.43 | 17.51 | 0 | 27.75 | 36.9 |
| Yi-6B-200K | 3.5 | 0 | 4.29 | 0.51 | 0.75 | 22.15 | 32.54 |
| ChatGLM3-6B-128K | 6.5 | 0 | 8 | 5.33 | 1 | 25.58 | 36.57 |
| **MiniCPM-2.4B-128K** | 9.5 | 0 | 4.29 | 22.08 | 0 | 27.32 | 37.68 |

Table 8: MiniCPM-2.4B-128K result in ∞Bench (Zhang et al., 2024b)

## I.3 MiniCPM-MoE

We further extend the ability of MiniCPM using Mixture-of-Expert.

**Initialization.** MiniCPM-MoE is initialized utilizing Sparse Upcycling (Komatsuzaki et al., 2022). The dense model checkpoint, derived from the stable phase of MiniCPM, undergoes a transformation wherein each MLP layer is substituted by an MoE layer. These new MoE layers are exact replicas of the original MLP layers from the dense checkpoint. The router parameters are randomly initialized following a normal distribution with a mean of 0 and a variance of 0.01.

**Routing Mechanism.** The number of total non-embedding parameters of MiniCPM-MoE is 13.6B. During training and inference, two out of eight experts are activated for each token, resulting in the number of activated parameters being approximately 4B. To prevent training from collapsing, an additional load balancing loss (Fedus et al., 2022) is applied to the final training objective. This auxiliary loss is multiplied by 0.01 which is large enough to ensure a balanced distribution of tokens assigned to different experts.

**Training.** Similar to MiniCPM, we employ WSD as our learning rate scheduler. Regarding the training data, we adhere strictly to the distribution specified in Section 6.2. The training batch size is maintained at 4M tokens during the stable training and decay stages and is reduced to 2M tokens during the SFT stage. The pre-training phase (including continue pre-train and decay stage) spans 130K steps, after which we notice diminishing improvement. The benchmark results are detailed in Table 9.

| Model | C-Eval | CMMLU | MMLU | HumanEval | MBPP | GSM8K | MATH | BBH |
|---|---|---|---|---|---|---|---|---|
| Llama2-34B | - | - | 62.6 | 22.6 | 33.0$^\dagger$ | 42.2 | 6.24 | **44.1** |
| Deepseek-MoE (16B) | 40.6 | 42.5 | 45.0 | 26.8 | 39.2 | 18.8 | 4.3 | - |
| Mistral-7B | 46.12 | 42.96 | **62.69** | 27.44 | 45.20 | 33.13 | 5.0 | 41.06 |
| Gemma-7B | 42.57 | 44.20 | 60.83 | 38.41 | 50.12 | 47.31 | 6.18 | 39.19 |
| MiniCPM-2.4B | 51.13 | 51.07 | 53.46 | 50.00 | 47.31 | 53.83 | 10.24 | 36.87 |
| **MiniCPM-MoE (13.6B)** | **58.11** | **58.80** | 58.90 | **56.71** | **51.05** | **61.56** | **10.52** | 39.22 |

Table 9: Benchmark results of MiniCPM-MoE. $^\dagger$ means evaluation results on the full set of MBPP, instead of the hand-verified set (Austin et al., 2021). The evaluation results of Llama2-34B and Qwen1.5-7B are taken from their technical reports.

## J  Edge Device Benchmarking

After Int4 quantization in Appendix H, MiniCPM-2.4B's footprint is reduced to 2GB, facilitating deployment on mobile edge devices. We adapted the model for Android and HarmonyOS using MLC-LLM (team, 2023b) and employed LLMFarm (team, 2023a) for adaptation to the iPhone system. This adaptation was tested on various edge mobile devices.

It is important to emphasize that our efforts did not focus on optimization for mobile deployment, but rather on demonstrating the practicality of MiniCPM's inference capabilities on mobile platforms. We encourage further optimization and updates by the developer community to enhance the performance of large models like MiniCPM in mobile contexts.

The result is shown in Table 10, we can see that on the most advanced smartphone iPhone 15 Pro, the inference throughput is as high as 18 tokens per second. In other devices, the inference throughput is also acceptable.

## K  Case Demonstration

A notable feature distinguishing the MiniCPM series from other robust SLMs like Phi (Li et al., 2023b) by Microsoft is that we train MiniCPM on general-purpose corpora, which guarantees the models' versatility and generalization across a variety of tasks. In this section, we present several intriguing generation instances, typically achievable only by larger models, to demonstrate the capabilities of MiniCPM. Although we do not conduct extensive checks on whether specific cases exist in the training set, due to the diverse ways they can be expressed, we ensure that no data similar to the test cases is intentionally added.

| SmartPhone | Operating System | Processor | Phone Memory (GB) | Inference Throughput (token/s) |
|---|---|---|---|---|
| OPPO Find N3 | Android 13 | snapdragon 8 Gen2 | 12 | 6.5 |
| Samsung S23 Ultra | Android 14 | snapdragon 8 Gen2 | 12 | 6.4 |
| Meizu M182Q | Android 11 | snapdragon 888Plus | 8 | 3.7 |
| Xiaomi 12 Pro | Android 13 | snapdragon 8 Gen1 | 8+3 | 3.7 |
| Xiaomi Redmi K40 | Android 11 | snapdragon 870 | 8 | 3.5 |
| Oneplus LE 2100 | Android 13 | snapdragon 870 | 12 | 3.5 |
| Oneplus HD1900 | Android 11 | snapdragon 865 | 8 | 3.2 |
| Oneplus HD1900 | Android 11 | snapdragon 855 | 8 | 3.0 |
| Oneplus HD1905 | Android 10 | snapdragon 855 | 8 | 3.0 |
| Oneplus HD1900 | Android 11 | snapdragon 855 | 8 | 3.0 |
| Xiaomi MI 8 | Android 9 | snapdragon 845 | 6 | 2.3 |
| Huawei Nova 11SE | Harmony 4.0.0 | snapdragon 778 | 12 | 1.9 |
| Xiaomi MIX 2 | Android 9 | snapdragon 835 | 6 | 1.3 |
| iPhone 15 Pro | iOS 17.2.1 | A17 pro | 8 | 18.0 |
| iPhone 15 | iOS 17.2.1 | A16 | 6 | 15.0 |
| iPhone 12 Pro | iOS 16.5.1 | A14 | 6 | 5.8 |
| iPhone 12 | iOS 17.2.1 | A14 | 4 | 5.8 |
| iPhone 11 | iOS 16.6 | A13 | 4 | 4.6 |

Table 10: Speed of MiniCPM-2.4B on different end devices.

---

**Translation**

**Prompt:** Translate this sentence into French: "I am a freshman on Chinese, do you know how this sentence is translated: 如何用Python创建一个简单的网页爬虫？" (How to create a simple web crawler in Python?)

**MiniCPM-1.2B:**
Je suis un homme nouveau en chinois, vous êtes-vous au courant de la manière dont cette phrase est traduite : Comment créer un petit scraper web simple à Python ?

**MiniCPM-2.4B-DPO:**
Je suis un nouveau-né en chinois, sais-tu comment on crée un simple web crawler en Python ? Translation: "Je suis un nouveau-né en chinois, sais-tu comment on crée un simple web crawler en Python ?"

Figure 19: Examples on translation. We notice that MiniCPM-2.4B-DPO performs better in translating the technical part, but both models are unnatural in expressing "just starting to learn Chinese.". However, both models successfully understand the instruction with traps and generate French despite we do not include French corpus intentionally.

**Math and Reasoning**

**Prompt:** 如果以下算式中减号均表示加号，而加号实际上是减号，给出算式的计算结果15-25-12+34=

(If in the following equation, minus signs represent addition, and plus signs represent subtraction, give the result of the calculation: 15-25-12+34=)

**MiniCPM-1.2B:**
如果减号表示加号，那么我们可以将算式改写为：15 + 25 + 12 - 34

首先计算前两个数的和：15 + 25 = 40
然后计算前三个数的和：40 + 12 = 52
最后计算最后一个数减去前三个数的和：52 - 34 = 18

所以，算式的计算结果为18。
(If the minus sign represents a plus sign, then we can rewrite the equation as: 15 + 25 + 12 - 34.

First, calculate the sum of the first two numbers: 15 + 25 = 40.
Then, calculate the sum of the first three numbers: 40 + 12 = 52.
Finally, calculate the difference between the last number and the sum of the first three numbers: 52 - 34 = 18.

Therefore, the result of the equation is 18.)

**MiniCPM-2.4B-DPO:**
如果减号均表示加号，而加号实际上是减号，那么算式可以重新表示为：
15+25+12-34

现在我们可以按照加减法的顺序进行计算：
(15+25)+12-34=40+12-34=52-34=18

所以，最终的计算结果是18。
(If minus signs all represent addition, and plus signs actually represent subtraction, then the equation can be re-expressed as:
15+25+12-34.

Now we can calculate according to the order of addition and subtraction:
(15+25)+12-34 = 40+12-34 = 52-34 = 18

So, the final result of the calculation is 18.)

Figure 20: Examples on math and reasoning. Both models are successful in symbolic substitution and math calculation.

**Instruction Following**

**Prompt:** 写藏头诗，藏头是龙年大吉

(Writing an acrostic poem, with the hidden message "Good luck in the Year of the Dragon.")

**MiniCPM-1.2B:**
龙池春水涨，
凤阙日初长。
吉庆何时至，
太平有象藏。
(The spring water in the dragon's pool rises,
The sun rises at the phoenix gate.
When will good fortune arrive,
Peace and prosperity are hidden in signs.)

**MiniCPM-2.4B-DPO:**
龙腾万里气如虹，
年岁新来乐事浓。
大吉良辰堪共赏，
吉祥如意笑颜红。
(Dragons soar for thousands of miles, their energy like a rainbow,
As the new year begins, joyous occasions abound.
Auspicious days are worth enjoying together,
Good fortune and happiness bring rosy smiles.)

Figure 21: Examples on instruction following. MiniCPM-2.4B-DPO successfully follows the instructions and generates a poem with the correct tonal pattern and the first character of each sentence. However, MiniCPM-1.2B only generates the correct tonal pattern.

---

**Emoji**

**Prompt:** 请给我一个json，每个key是一个emoji，每个value是这个emoji对应的意思，给出10个左右

(Please give me a JSON object where each key is an emoji and each value is the meaning of that emoji, providing around 10 examples.)

**MiniCPM-1.2B:**

当然可以，这是一个包含10个左右emoji及其含义的JSON：

{

"😊": "笑容",

"😂": "笑",

"😍": "惊叹",

"😘": "亲吻",

"😜": "恶作剧",

"😔": "失望",

"🤔":"思考",

"🤑": "金钱",

"🙏": "帮助",

"💩": "废物"

}

希望这个JSON对你有帮助！

(Of course, this is a JSON file containing around 10 emojis and their meanings:

{

"😊": "smiling",

"😂": "laughing",

"😍": "amazing",

"😘": "kissing",

"😜": "mischief",

"😔": "disappointment",

"🤔":"thinking",

"🤑": "money",

"🙏": "help",

"💩": "rubbish"

}

Hope you found this JSON useful!)

**MiniCPM-2.4B-DPO:**

当然可以！以下是一个包含10个emoi及其对应意义的JSON对象：

{

"😊": "表示愉快或满意，通常用于表达微笑和喜悦。",

"😠": "表示愤怒或不满，通常用于表达情绪的激动。",

"😂": "表示大笑或开心，通常用于表达笑声和欢乐。",

"🤣": "表示搞笑或尴尬，通常用于表达搞笑的场景或尴尬的情况。",

"🤯": "表示惊讶或震惊，通常用于表达对某个事件或情的强烈反应。",

"😔": "表示失望或沮丧，通常用于表达对某事或某人失去兴趣或信心的情绪。",

"💔":"表示心痛或伤心，通常用于表达对某事或某人造成痛苦的强烈感受。",

"💡": "表示惊喜或兴奋，通常用于表达对某事或某人的发现或发现的情绪。",

"😎": "表示酷或时髦，通常用于表达自信、独立和潮流的态度。",

"😍": "表示喜欢或欣赏，通常用于表达对某物或某人的喜爱之情。"

}

希望这个JSON对象能满足您的需求！如果您有任何问题，请随时告诉我。

(Of course! Here is a JSON object containing 10 emojis and their corresponding meanings:

{

"😊": "It means happy or satisfied, typically used to express smiling and joy.",

"😠": "It means anger or dissatisfaction, typically used to express intense emotions.",

"😂": "It means laughing or happy, typically used to express laughter and joy.",

"🤣": "It means funny or awkward, typically used to express humorous situations or awkwardness.",

"🤯": "It means surprise or shock, typically used to express a strong reaction to an event or situation.",

"😔": "It means disappointment or discouragement, typically used to express a loss of interest or confidence in something or someone.",

"💔": "It means heartache or sadness, typically used to express intense feelings of pain caused by something or someone.",

"💡": "It means surprise or excitement, typically used to express the emotion of discovery or finding something or someone.",

"😎": "It means cool or stylish, typically used to express a confident, independent, and trendy attitude.",

"😍": "It means like or appreciate, typically used to express affection or admiration for something or someone."

}

I hope this JSON object meets your needs! If you have any questions, please feel free to let me know.)

Figure 22: Examples on emoji. Both models understand emojis and structured data well, and accurately control the number of items implicitly. The DPO model tends to generate more words.

