# OpenReview forum: "MiniCPM: Unveiling the Potential of Small Language Models with Scalable Training Strategies"
_colmweb.org/COLM/2024/Conference — COLM_

### Official Review · Reviewer_S8K9 · 2024-04-23

**Rating:** 9
**Confidence:** 4
**Ethics Flag:** 1

**Summary:**

The manuscript presents a detailed study on the development and application of MiniCPM, a series of small language models (SLMs), which demonstrate comparable capabilities to much larger language models (LLMs) in the range of 7B-13B parameters. The authors have addressed a timely topic in the machine learning community, focusing on resource efficiency and practical utility of SLMs.

**Questions To Authors:**

1) What is the total model size when embedding parameters are included?
2) The manuscript lacks a discussion on the utilization of hardware resources, such as GPUs. Could you provide details on this aspect?

**Reasons To Accept:**

1) The paper is well-motivated, addressing both the economic and practical challenges of deploying large models.
2) The MiniCPM models are robustly presented, with detailed descriptions of the architecture, training methods, and innovative approaches like the WSD learning rate scheduler.
3) The experimental section is comprehensive, demonstrating the models' performance across various benchmarks, which substantiates the authors' claims.

**Reasons To Reject:**

1) Does the emergence pattern of Small Language Models (SLMs) mirror that of Large Language Models (LLMs)? If there are differences, can the insights gained from SLMs still be applicable for the development of LLMs?
2) The recent advancements with models like Llama-3 suggest that smaller-scale models can achieve enhanced performance with increased training data. Does this paper offer any additional insights or conclusions in this regard?

---

> ### Author Rebuttal · Authors · 2024-05-31
>
> Thanks for the appreciation of our work! We would like to clear up the confusion further you raised.
>
> - Emergence Pattern
>   - We assert that emergence may manifest similarly across both the model and data scales. More fundamentally, emergence patterns may be influenced by factors such as compute resources or pretraining loss/PPL [1]. This implies that a relatively SLM trained with extensive data might exhibit a comparable emergence pattern to that of an LLM trained with less data. Nevertheless, we acknowledge this as an insightful query: Is there any inherent difference in emergence patterns?  This question is worthy of future study.
>     - [1] Understanding Emergent Abilities of Language Models from the Loss Perspective
> - Insight transferability
>   - Regarding the "insight gained from SLMs," our strategies are intentionally designed with scalability in mind, encompassing both model-wise scaling and data-wise scaling. Although our current resources restrict us from training a larger model, we firmly believe that the strategies presented in our paper can be effortlessly extended to train much larger models and applied to other models as well. In fact, we have observed several recent efforts that successfully implement our method in such a short time, underscoring its broader applicability. To name a few:
>
> ```
> [1] Shen et al. JetMoE: Reaching Llama2 Performance with 0.1M Dollars https://arxiv.org/abs/2404.07413
> [2] Light-on.  Passing the Torch: Training a Mamba Model for Smooth Handover. https://www.lighton.ai/blog/lighton-s-blog-4/passing-the-torch-training-a-mamba-model-for-smooth-handover-54
> [3] Elie@Huggingface https://x.com/eliebakouch/status/1790772100585722088
> [4] Glorioso et al. Zamba: A Compact 7B SSM Hybrid Model. https://arxiv.org/pdf/2405.16712
> [5] MAP-Neo: Highly Capable and Transparent Bilingual Large Language Model Series https://arxiv.org/pdf/2405.19327
> ```
> - Question about LLama3
>   - In a recent revision of this paper, we carefully measured the scaling law for our model. We found that current models can "absorb" much more data than the Chinchilla-Optimal amount. Specifically, the compute-to-data ratio for our model is now greater than 200. This finding is in line with the trend observed in models like Llama3 (adding more data is helpful).
> - Parameters count
>   - 2.7B parameters with embeddings.
> - Training resources
>   - For MiniCPM-2.4B itself, we utilize 160 A100 GPUs trained for 2 weeks (do not count in the preliminary studies).

---

> > ### Author Response · Authors · 2024-05-31
> > **Reference updating**
> >
> > [3] was updated into a paper very recently:  Scaling Laws and Compute-Optimal Training Beyond Fixed Training Durations https://arxiv.org/pdf/2405.18392

---

> ### Comment · Area_Chair_vPnZ · 2024-06-05
> **Please respond to rebuttal and other reviewers' comments**
>
> Reviewer S8K9, could you please reply and/or update your review given the authors' response and discussion with the other reviewers? Thanks!

---

### Official Review · Reviewer_PFXg · 2024-04-29

**Rating:** 7
**Confidence:** 4
**Ethics Flag:** 1

**Summary:**

This paper investigates the capabilities of small language models (approximately 2 billion parameters), a critical area in current language model research. It introduces MiniCPM, a family of models with 1.2B and 2.4B parameters that exhibit competitive performance. This success is attributed to a thorough hyper-parameter scaling analysis on smaller-scale models and a novel training strategy employing a learning rate scheduler named Warmup-Stable-Decay. While the paper is generally well-written and provides detailed empirical results and discussions, some issues with notation clarity and figure presentation persist, and additional ablation studies are recommended to validate the effectiveness of the proposed methods.

**Questions To Authors:**

- Are optimal batch size and optimal learning rate independent? Analyses in Section 3.2 and 3.3 seem to be done separately.
- Will the training code, including the implementation of WSD, be made publicly available to facilitate reproducibility in research?

**Reasons To Accept:**

- Exploring the capabilities of small language models is significantly valuable for advancing research in both small and large language models.
- The paper presents a comprehensive and effective training strategy for small language models, alongside clear decision-making insights derived from empirical results. These contributions are likely to inspire further research in the field.
- The proposed model family shows impressive performance across a variety of benchmarks, even when compared to much larger models.

**Reasons To Reject:**

- The paper contains ambiguities and possible errors in its notation, which impede understanding. For instance:
  - This paper uses notations like "WSD(80N,8N)'' multiple times. However, there is no definition about what the two items in parentheses represent.  $WSD(T,D)$ in Appendix B.1 is also not defined.
  - In the "Decay Stage" section, the equation for exponential annealing $f(s-T)=\eta \times 0.5^{(s-S)/T}$ appears to be incorrect. The variable $\eta$ seems out of place, and when $s=T$, the learning rate should equal $\eta$.
- Several figures are poorly presented:
  - The labels in Figures 1 and 2 are too small to read.
  - It is difficult to distinguish between the colors of different lines in Figures 5 and 6.
  - Despite descriptions suggesting the presence of a curve showing the 0.036B model (with 4 times compute) matching the performance of the 0.17B model with WSD (40N, 4N), this curve is absent from Figure 6.
- This paper highlights the benefits of introducing high-quality SFT data at the beginning of annealing. However, mixing SFT data with pretraining data after the pretraining could also improve the performance. That is to say, what may be important is the data mixing operation rather than the training stage. The ablation study did not shows whether mixing SFT data at the beginning of annealing is the best choice.
- Most baselines in Table 2 do not use SFT, whereas MiniCPM is trained with both pretraining and SFT data. Additional discussion or experiments are needed to determine how much of MiniCPM’s success is due to the inclusion of SFT data.

---

> ### Author Rebuttal · Authors · 2024-05-31
>
> Sorry about the confusion, we will make them clear as much as possible in the revision.
> - Ambiguity.
>   -  Despite Equation 1 might help to understand WSD(80N, 8N), we will make further clarification as follows (and will be updated in the next revision):  WSD(T; S) means stable training on T tokens, and decay on S tokens, where T, S are usually represented relative to the model size N, i.e., KN tokens means the number of tokens is K times the number of parameters of the model.
> - Incorrect formula. Thanks for the careful review. We have revised the formula in our revision: $f(s-T) = 0.5^{(s-S)/T}$
> - Unclear Figures
>   - We have revised the font size in our current revision.
>   - We will revise the color of the lines in our next revision.
> - Curve missing
>   - This curve is present in Figure 6. In Figure 6, we fit the optimal loss envelope for continuously training the 0.036B model. At $2.5\times 10^{19}$ compute, the curve arrives at a 3.32 loss. The 0.17B model also arrives at a 3.32 loss with about $0.6\times 10^{19}$ Flops. This information results in the conclusion you mentioned. We did not train the 0.036B models to $2.5\times 10^{19}$ Flops in practice.
> - Ablation about the training stage
>   - We include the comparison that helps to show "mixing SFT data at the beginning of annealing" in Appendix C, Table 5. Where the models trained on SFT data but not at the beginning of annealing lag behind our proposed strategy significantly. We hope this addresses your concern.
> - Ablation of data mixture
>   - The success of MiniCPM undeniably involves the integral of SFT high-quality data. We have attempted annealing without any SFT data, and don't do SFT either, and the results, unsurprisingly, were unsatisfactory (e.g., MMLU~26). However, it is quite challenging to perform ablation studies on individual SFT datasets. We intend to leave the work on data mixture and data quality for future research.
> - optimal batch size and optimal learning rate
>   - They are likely to be not independent. To overcome this correlation, we do a preliminary study on the LR first, then choose an optimal learning rate to do a batch size experiment, and use batch size scaling to do the LR again. It is a bit like the Coordinate Descent optimization method. We will add this comment in the revision
> - code availability
>   - The training code is tightly integrated with in-house infra, which is not suitable for open source. However, the code of WSD and inference code are open-sourced publicly.

---

> > ### Comment · Reviewer_PFXg · 2024-06-05
> >
> > I appreciate the author's response, which answers most of my concerns. I hope to see these changes in the updated version. I will increase my rating score.

---

> > > ### Author Response · Authors · 2024-06-05
> > >
> > > Thanks! We are updating the paper! You will see the changes in both future public versions and the camera-ready version if accepted.

---

### Official Review · Reviewer_qVU8 · 2024-05-10

**Rating:** 7
**Confidence:** 3
**Ethics Flag:** 1

**Summary:**

In this paper, the authors study the training dynamics of small language models (i.e., models ranging from 1.2-2.4 billion parameters).  Specifically, they focus on what they call “model wind tunnel” experiments. This involves training and studying model developing in 3 parts: 1) hyper-parameter selection (mostly focusing on network width and depth scaling parameters); 2) optimal batch sizing and; 3) learning rate scheduling. The most novel part of their work seems to be in the last part related to the development of what they call “warmup stable decay” (WSD), their learning rating scheduler that is conducive to continuous training and training without a fixed token length.

These experiments above lead to the following results (centering around experiments on C4): a new empirical bound on the optimal relationship between batch size and C4 loss that helps to minimize both token amount and loss (reported at the end of Section 3.2). Compelling empirical results on the efficacy of their WSD scheduling method (which breaks the scheduler into 3 stages: warmup, stable and decay stages, informed by systematic experiments they performed on cosine-warmup and its sensitivity to various step parameters); in particular, they report impressive losses with very small models on C4 (Figures 5 and 6).

Based on these insights, they then train a family of small language models called miniCPM. Their training pipeline is comprehensive, ranging from full pre-training (at the scale of trillion of tokens using aggregations of multilingual publicly available pre-training corpora), to supervised instruction tuning and DPO-style alignment. They also tested across different architectures (e.g., MoE derivates and long-context versions of their models). The main novelty in this pre-training setting is the introduction of SFT data into the decay stage of their pre-training and scheduler (while I know this has become popular d recently, I worry that this makes their models have an unfair advantage over some of the other models they are comparing against; more about this below).

The results are impressive and, after looking through the details in the appendix, I was happy to see that not only were all the model comparisons produced by the authors themselves using a single tool, but that performance reported is based on each model’s best performance either through perplexity-based evaluation or direct generation. While I initially found Table 2 to be a bit confusing (highlighting would be very helpful here!), I was particularly struck by the relative average performance of 13-40B models against MiniCPM-2.4B.

**Questions To Authors:**

- Will you models be released, and if so under what license?
- I was surprised not to see (unless I missed it) details about the the particular optimizer you used. Can you clarify this?
-  I was surprised to that your SFT stage includes training on 6 billion tokens. Did you report what sources you used?
- you called the alignment stage RLHF, but as far as I can tell, you weren't using RL, because you were training with DPO, right? Please clarify.

other comments

- The caption is Table 1 is very hard to read, please revise.

**Reasons To Accept:**

- A very careful study on optimal model training strategies for small LLMs, which is an important area of research that, as the authors convincingly argue, touches on important issues related to practical deployment of LMs.

- The paper is very easy to read and I found all the experiments to be well motivated. I believe that their experiments will motivate further research in this area.

- A new suite of small LMs that I think will be used and embraced by the community (the question of whether these models will be released was not addressed; see questions below).

**Reasons To Reject:**

- My main concern centers around the introduction of instruction data into the pre-training stage, and whether this makes the comparisons fair to many of the models in Table 2 (models that presumably didn't do this). While I don't think that this is grounds for a rejection, I would like to see the authors motivate and address this. If there were any ablations on this (maybe that I would missed), it would be helpful to report.  (Given that this was done in the most recent OLMO 1.7 model, this might be worthwhile to compare against)

---

> ### Author Rebuttal · Authors · 2024-05-31
>
> Thanks for the appreciation of our work. We apologize for any omissions in the initial submission and will address all the concerns in our revision to ensure clarity and accuracy.
> - Introduction of SFT Data into the Decay Stage
>   - Thank you for pointing this out. We believe that the comparison is not unfair due to the inclusion of SFT data during the decay stage. Current SOTA models often blur the boundaries between raw pretraining data and high-quality, human-curated data. As most models do not disclose their training data, it cannot be conclusively said that other models did not include such high-quality data. Given that most SOTA models release both base models and Chat models, we adhere to this convention. With the increasingly blurred boundaries between human-curated data and web-crawled data, we argue that directly comparing the SFT-ed models ensures a fairer comparison.
>   - We acknowledge that the comparison between Falcon-40B/MPT-30B and our model might be somewhat unfair concerning data differences. At the time of their training, it was less likely that they included high-quality data now being introduced. This comparison underscores that the advancements in pretraining techniques have significantly boosted model performance.
> - Model and License
>   - The model has been fully released, including intermediate checkpoints, under the Apache-2.0 license. However, due to anonymity constraints, I cannot directly point you to these resources.
> - Optimizer
>   - We use the Adam optimizer, and we will specify this in the revised manuscript.
> - SFT Data Mixture
>   - The data distribution for the decay stage, which includes the SFT data, is detailed in Appendix D.2, specifically in Figure 13. The SFT data consists of EvolInstruct, OssInstruct, SlimOrca, ShareGPT4, UltraChat, and a substantial portion of in-house SFT data covering logic, code, math, and knowledge.
> - Use "Alignment" Instead of "RLHF"
>   - You are absolutely correct. DPO is not RLHF, and we will make this correction in the revision.
> - Table 1 Unclear
>   - We will revise Table 1 to ensure it is clearer. Is there any particular point in Table 1 that you found especially unclear? Please let us know so we can specifically address it.
>
> *Additionally, we list some of the recent implementations of our method in other powerful models (See our comments to the last reviewer S8K9). We are impressed by the community's love of our training strategy and the wide spread of it in such a short time.*

---

> ### Comment · Reviewer_qVU8 · 2024-06-03
>
> Thank you for your feedback. I think you make fair points, especially for the first bullet point. I will keep my score as it is for now.

---

### Official Review · Reviewer_n21G · 2024-05-11

**Rating:** 7
**Confidence:** 3
**Ethics Flag:** 1

**Summary:**

This work presents a suite of models called the MiniCPM models, which are language models of moderate size (~1.2B and ~2.4B parameters) that perform comparably to some of the strongest 7B models that are available (incl. Mistral-7B). Their main contribution is introducing a warmup-stable-decay (WSD) learning rate scheduler and analyze properties of this. They integrate WSD into their system and enjoy performance bumps. Finally they contribute an MoE model and long-context model based on the MiniCPM architecture.

**Questions To Authors:**

Clarity things

1) N throughout is number of parameters of the model right? When you say S = 20N for a 1B model this would be 20B tokens of training, right? It would be great to make this more clear in the body of the text.

2) The paper was hard to follow and there are a lot of awkward and/or redundant phrasing that made things unclear (e.g. in the batch size section 'the same number of GPUs is kept constant').

**Reasons To Accept:**

1) WSD seems like quite a simple approach and can be integrated fairly easily into LM pretraining.

2) Performance seems good in general, being on par with or close to models 3-4x the size.

3) Analysis of the hyperparameters to get something that ends up working is good.

4) Releasing a suite of models including an MoE, long context, and potentially a vision one (this was stated to maybe be a part of an upcoming paper, but also slated in here, which confused me).

**Reasons To Reject:**

1) Although it seems like WSD works well on very small scales that you've tested, it would've been nicer to see how the exact same model with the exact same data order without WSD would do across the board. I think this could be computationally prohibitive, but that baseline would be nice to see.

2) It seems like nearly all of the contribution here is the algorithm for WSD, but that's conflated with the fact that you're releasing a set of performant models, so its a little distracting and poorly organized. (minor point)

3) There are some clarity issues that I expand a little bit more about in the questions to the authors. (minor point)

---

> ### Author Rebuttal · Authors · 2024-05-31
>
> Thanks for the careful review and appreciation of MiniCPM. We are sorry about the confusion, and we will revise the paper carefully according to your suggestion.
> - More Careful Comparison Between WSD and Cosine
>     - We show that WSD can perform on par with cosine on loss on smaller models at any data scale, which already serves as an important reason for switching to the WSD scheduler. However, we can not immediately provide a very rigorous comparison when the training data is very large in the current version. We will leave it for future work.
> - Conflation of Contributions
>     - Thanks for pointing that out. Our paper mainly introduces mainly MiniCPM base models including their training strategy (WSD, hyper-parameter, two-stage training). These elements collectively contribute to the model's performance.
>     - We do think WSD is the most novel part. In the revision, we will make this core contribution clearer.
>     - For the other models, we mainly introduce them in the appendix (optional for reading). For these parts, it is more like a technical report that acknowledges all project members of our organization and also benefits relevant researchers.
>     - As for the vision part, we will remove the mention in the next revision. (During the submission of this work, we have not decided on whether to publish the vision paper).
> - Notation N's Meaning
>     - Yes, your understanding is correct. We will make it clearer in the next revision.
> - Writing Needs Improvement
>     - Thanks for pointing out the clear phrases. We have continued polishing the paper since our submission, and these improvements will be presented in the next revision.
>
> *Additionally, we list some of the recent implementations of our method in other powerful models as follows: (also see our comments to the last reviewer S8K9). We are impressed by the community's love of our method and the wide spread of it in such a short time.*
> ```
> [1] Shen et al. JetMoE: Reaching Llama2 Performance with 0.1M Dollars https://arxiv.org/abs/2404.07413
> [2] Light-on.  Passing the Torch: Training a Mamba Model for Smooth Handover. https://www.lighton.ai/blog/lighton-s-blog-4/passing-the-torch-training-a-mamba-model-for-smooth-handover-54
> [3] Elie@Huggingface https://x.com/eliebakouch/status/1790772100585722088
> [4] Glorioso et al. Zamba: A Compact 7B SSM Hybrid Model. https://arxiv.org/pdf/2405.16712
> [5] MAP-Neo: Highly Capable and Transparent Bilingual Large Language Model Series https://arxiv.org/pdf/2405.19327
> ```

---

> > ### Comment · Reviewer_n21G · 2024-06-05
> > **Response to the rebuttal**
> >
> > Thanks for answering the questions and concerns I have. I think I under-valued the demonstrated utility of WSD, so after considering some of these things I'm inclined to raise my score to a 7. I'm updating the review to reflect this.

---

> > > ### Author Response · Authors · 2024-06-05
> > >
> > > Thanks for your appreciation! We hope more people know MiniCPM series and use WSD in their practice.

---

### Decision · Program_Chairs · 2024-07-10

**Decision:**

Accept

**Comment:**

In this paper the authors present a family of smaller LLMs in the 1.2-2.4B range, MiniCPM, that perform comparably to 7-11B parameter models. Their main contribution is the methodology for obtaining such performance using so many fewer parameters, which is achieved primarily though a new staged learning rate scheduler, warmup-stable-decay (WSD), designed for the continual training/adaptation setting, and a corresponding training recipe. The models have been released, including intermediate checkpoints, under the Apache-2.0 license.

Reviewers agreed that the paper is reasonably well written and represents an interesting and impactful contribution, describing a new training recipe that allows for LLMs to perform competitively with respect to models that are 3-4x their size.